# The olfactory receptor SNIF-1 mediates foraging for leucine-enriched diets in *C. elegans*

Ritika Siddiqui[1†], Nikita Mehta[1†], Gopika Ranjith[2†], Marie-Anne Félix[3], Changchun Chen[4], Varsha Singh[1,2*]

[1]Division of Molecular Microbiology, School of Life Sciences, University of Dundee, Dundee, United Kingdom; [2]Department of Developmental Biology and Genetics, Indian Institute of Science, Bangalore, India; [3]Institute of Biology of the Ecole Normale Supérieure, Paris, France; [4]Department of Molecular Biology, Umeå University, Umeå, Sweden

## eLife assessment

This **important** work is the first to suggest a model that the nematode *C. elegans* prefers specific bacteria (its major food source) that release high amounts of the known attractant isoamyl alcohol when supplemented with exogenous leucine and has also identified a likely receptor for the odorant isoamyl alcohol. The evidence supporting the claims of the authors is **solid**, and the manuscript would be improved by changes to the text that clarify and address the distinction between "supplemented" versus "enriched". The renaming of srd-12 to snif-1 should also be addressed.

**\*For correspondence:**
vsingh001@dundee.ac.uk

[†]These authors contributed equally to this work

**Competing interest:** The authors declare that no competing interests exist.

**Abstract** Acquisition of essential nutrients through diet is crucial for the survival of animals. Dietary odors might enable animals to forage for nutrient-rich diets. We asked if *Caenorhabditis elegans*, a bacterivorous nematode, uses olfactory cues to forage for essential amino acid-rich (EAA) diets. Using the native microbiota of *C. elegans,* we show that worms rely on olfaction to select leucine (EAA)-supplemented bacteria. Using gas chromatography, we find that leucine-supplemented bacteria produce isoamyl alcohol (IAA) odor in the highest abundance. Prior adaptation of worms to IAA diminishes the diet preference of worms. Several wild isolates of *C. elegans* display robust responses to IAA, emphasizing its ecological relevance. We find that foraging for a leucine-supplemented diet is mediated via the AWC olfactory neurons. Finally, we identify SNIF-1 G protein-coupled receptor in AWC neurons as a receptor for IAA and a mediator of dietary decisions in worms. Our study identifies a receptor-ligand module underpinning foraging behavior in *C. elegans*.

## Introduction

Amino acids are central to metabolism, not only as building blocks for proteins but also as signaling molecules. During the course of evolution, animals have lost the ability to synthesize half of the proteogenic amino acids, making dietary intake of such amino acids indispensable (*Gietzen and Rogers, 2006*). Leucine (Leu), isoleucine (Ile), valine (Val), histidine (His), lysine (Lys), tryptophan (Trp), phenylalanine (Phe), methionine (Met), threonine (Thr), and arginine (Arg) constitute essential amino acids (EAAs) (*Tomlinson et al., 2011*). Several studies have investigated the mechanism of amino acid surveillance by cells and tissues. Leucine, glutamine, and arginine have been shown to activate the mTOR pathway and control protein translation and cell growth (*Sancak et al., 2008*; *Hara et al.,*

*1998*; *van der Vos and Coffer, 2012*; *Bauchart-Thevret et al., 2010*; *Gauthier-Coles et al., 2021*; *Bar Peled and Sabatini, 2014*; *González and Hall, 2017*). Altered EAA levels influence the longevity, behavior, and health of animals (*Austad et al., 2024*; *Minussi et al., 2023*; *Enomoto et al., 2013*). Increased levels of circulating branched-chain amino acids (BCAAs) are associated with an increased risk of obesity and diabetes in humans (*Tai et al., 2010*; *Wang et al., 2011*). Restricted intake of tryptophan and methionine extends lifespan in rats, while a tryptophan-deficient diet reduces the weight of organs (*De Marte and Enesco, 1986*; *Miller et al., 2005*). Methionine restriction is reported to extend the lifespan of *Drosophila* (*Lee et al., 2014*). Central administration of a BCAA, leucine, reduces food intake in rats (*Cota et al., 2006*). An extended lifespan upon diet restriction in silkworms is linked to enhanced utilization of EAAs (*Wang et al., 2023*). However, it is not clear if animals have a way to modulate EAA-specific dietary intake. Do animals use specific sensing mechanisms to find an EAA-enriched diet? Several studies on feeding behaviors suggest that both invertebrates and vertebrates select nutrient-rich diets. For instance, rats prefer a protein-rich diet over a carbohydrate-rich diet (*Makarios-Lahham et al., 2004*). Many animals display mechanisms to choose for or reject food with specific EAAs. Sparrows consume a combination of different diets to get adequate levels of various EAAs, while gastropods reject diets deficient in methionine (*Delaney and Gelperin, 1986*; *Gietzen, 1993*; *Murphy and Pearcy, 1993*). When maintained on an EAA-restricted diet, flies compensate by increasing feeding (*Juricic et al., 2020*). Kittens prefer diets rich in methionine and threonine, while rats select for a leucine-enriched diet (*Rogers et al., 2004*). Despite evidence that organisms seek EAA, specific molecular mechanisms for search behavior are not well understood.

Chemoperception, taste and olfaction, via G-protein coupled receptors (GPCRs) allow animals to quickly assess their diet. EAAs evoke taste perceptions, such as sweet (Thr, His, Leu, Phe, Trp) and bitter (Arg, Ile, Lys, Met), while non-EAAs can elicit umami (Glu) and sour (Asp) taste (*Shallenberger, 1993*; *Schiffman et al., 1981*; *Zhao et al., 2016*). Amino acids that elicit sweet responses in humans, like threonine, are found to be attractive to rodents as well (*Bachmanov et al., 2016*; *Yoshida and Saito, 1969*). GPCRs are implicated in sensing different classes of amino acids and regulate foraging behavior in animals (*Kuang et al., 2003*; *Nelson et al., 2002*; *Conigrave et al., 2000*; *Conigrave and Hampson, 2006*). Metabotropic glutamate receptors (mGlu) sense glutamate, while the extracellular calcium-sensing receptor senses aliphatic, aromatic, and polar amino acids (*Conigrave et al., 2000*; *Nakanishi, 1992*). Invertebrates like *Caenorhabditis elegans* and *Drosophila melanogaster* also possess GPCR-based amino acid-sensing mechanisms (*Miguel-Aliaga, 2012*). *C. elegans* senses glutamate using mGlu type receptors, Mgl-1/2/3, while *D. melanogaster* utilizes a cationic amino-acid transporter, Dm *slif*, to sense arginine (*Dillon et al., 2015*; *Colombani et al., 2003*). The goldfish 5.24, an odorant receptor, responds to arginine, while its human homolog, GPRC6A, responds to basic as well as aliphatic amino acids (*Speca et al., 1999*; *Wellendorph et al., 2005*). Olfactory GPCRs are known to sense various odors, including alcohol, esters, ketones, etc., and many of these are products of EAA catabolism. It is not known if animals rely on EAA-derived odors to find an EAA-rich diet.

The presence of an elaborate chemosensory system in soil-dwelling animals suggests the usage of olfactory cues for foraging. *C. elegans,* a bacterivorous nematode, relies heavily on its simple but effective chemosensory system composed of 302 neurons for foraging, although specific cues regulating this phenomenon are unknown (*Sawin et al., 2000*; *Rhoades et al., 2019*). Olfactory neurons of *C. elegans* express as many as 335 GPCRs (*Hammarlund et al., 2018*); however, the odors stimulating them remain largely unknown. Like other invertebrates, *C. elegans* requires ten EAAs; of these, leucine, tryptophan, valine, arginine, and lysine extend their lifespan when supplemented at low concentration (*Edwards et al., 2015*). Does *C. elegans* rely on olfactory cues to sense diets with high(er) levels of EAAs? It is conceivable since some bacteria possess biosynthetic pathways to convert amino acids into odors, such as dimethyl sulfide, indole, isoamyl acetate, phenylethyl alcohol, and benzyl alcohol from methionine, tryptophan, leucine, and phenylalanine respectively (*Weisskopf et al., 2021*). *C. elegans* can sense hundreds of odors representing extensive structural diversity (*Bargmann et al., 1993*). Do worms utilize some of them as foraging cues to find nutrient-enriched bacteria?

We hypothesized that certain odors produced by the microbiota of *C. elegans* are olfactory foraging cues for EAA-rich bacteria and drive worms' foraging behavior. To test this hypothesis, we analyzed the preference of worms for natural microbiota species supplemented with individual EAAs. We found that odors drive worms' feeding preference in favor of leucine-supplemented diets. By carefully analyzing the odor bouquet produced by natural microbiota upon leucine supplementation,

we identified isoamyl alcohol (IAA) as the most abundant odor in preferred diets. We show that IAA is produced from leucine using the Ehrlich degradation pathway in bacteria. IAA is an ecologically relevant odor and regulates the preference of worms for a leucine-supplemented diet. Finally, we identified SNIF-1, a GPCR expressed in AWC neurons, as the receptor for IAA. Taken together, SNIF-1 regulates the dietary preference of worms to IAA-producing bacteria and thereby mediates the foraging behavior of *C. elegans* to leucine-supplemented diets. Thus, IAA produced by bacteria is a dietary quality code for leucine-supplemented bacteria.

## Results

### Microbial odors drive the preference of *C. elegans* for leucine-supplemented diets

To understand if *C. elegans* used olfactory cues to distinguish between standard and EAA-supplemented diets, we performed 'odor-only' diet preference assays using natural microbiota of *C. elegans*, CeMbio, and laboratory food, *Escherichia coli* OP50. CeMbio is a collection of 12 bacteria that best reflects the metabolic complexity of the microbes in the natural habitat of worms (see *Supplementary file 1*). N2 or wild-type (WT) worms were allowed to choose between odors from each strain grown on separate sections, EAA supplemented (+EAA) or unsupplemented (-EAA), of a tripartite plate (schematic in *Figure 1A*). Of the ten EAAs tested, WT worms showed a preference for all the bacteria supplemented with leucine (+LEU) over unsupplemented (-LEU) except for *Chryseobacterium scophthalmum* (JUb44) and *Escherichia coli* OP50 (*Figure 1B*). However, worms could not distinguish between supplemented and unsupplemented bacteria for the remaining nine EAAs for most of the diets (*Figure 1C, D*, *Figure 1—figure supplement 1A-G*). We found that worms did not respond to leucine at three different concentrations in a modified chemotaxis assay, suggesting that worms are incapable of sensing leucine directly (*Figure 1—figure supplement 1H*). These results suggested that worms can forage for diets supplemented with specific EAA, leucine, using odors produced by some bacteria.

We next asked whether worms prefer their native microbiota over laboratory food *E. coli* OP50. We performed 'odor-only' diet preference assays to test if worms can distinguish between CeMbio and *E. coli* OP50 using odors (schematic in *Figure 1E*). We found that the preference indices (PI) of worms for *Enterobacter hormaechei* (CEent1), *Lelliottia amnigena* (JUb66), and *Sphingobacterium multivorum* (BIGb0170) were positive (preferred diets), while neutral or negative for other strains, such as *Pseudomonas lurida* MYb11 (*Figure 1F*). Altogether, these findings suggested that worms rely on odors to distinguish various bacteria and find leucine-supplemented bacteria.

### Isoamyl alcohol odor is a signature for a leucine-supplemented diet

We hypothesized that bacterial odor(s) enriched upon leucine supplementation are likely to be drivers of foraging behavior in worms. We identified all the odors in the headspace of individual bacteria using solid-phase microextraction followed by gas chromatography-mass spectrometry (GC-MS/MS). By examining levels of individual odors in leucine-supplemented and unsupplemented bacteria, we found that several odors were present in the headspace of bacteria (*Figure 2A, B*, and *Figure 2—figure supplement 1A–D*, also see *Supplementary file 2*). However, isoamyl alcohol (IAA) was the most abundant and a shared odor in the worms' preferred diets, CEent1, JUb66, and BIGb0170 (*Figure 2A–C* and *Figure 2—figure supplement 1A–D*). We found that under standard conditions, IAA constituted 40–70% of the headspace of preferred bacteria and remarkably increased up to 90% upon leucine supplementation (*Figure 2C*). IAA was also detected in the headspace of some non-preferred bacterial strains (*Supplementary file 2*). We also quantified leucine-dependent increase in IAA levels in the headspace of preferred bacteria (*Figure 2—figure supplement 1E*). We found that the abundance of IAA significantly increased in CEent1, JUb66, and BIGb0170 up to fourfold upon leucine supplementation (*Figure 2D*, *Figure 2—figure supplement 1F, G*). To test if a fourfold increase in IAA concentration is sufficient to explain worms' preference to leucine-supplemented bacteria, we performed a chemotaxis assay where worms were given a choice between fourfold and onefold concentration of IAA (schematic in *Figure 2—figure supplement 1H*). We found that worms prefer higher concentration of IAA (*Figure 2—figure supplement 1I*). This suggested that worms can sense the gradient of IAA in chemically complex scenarios.

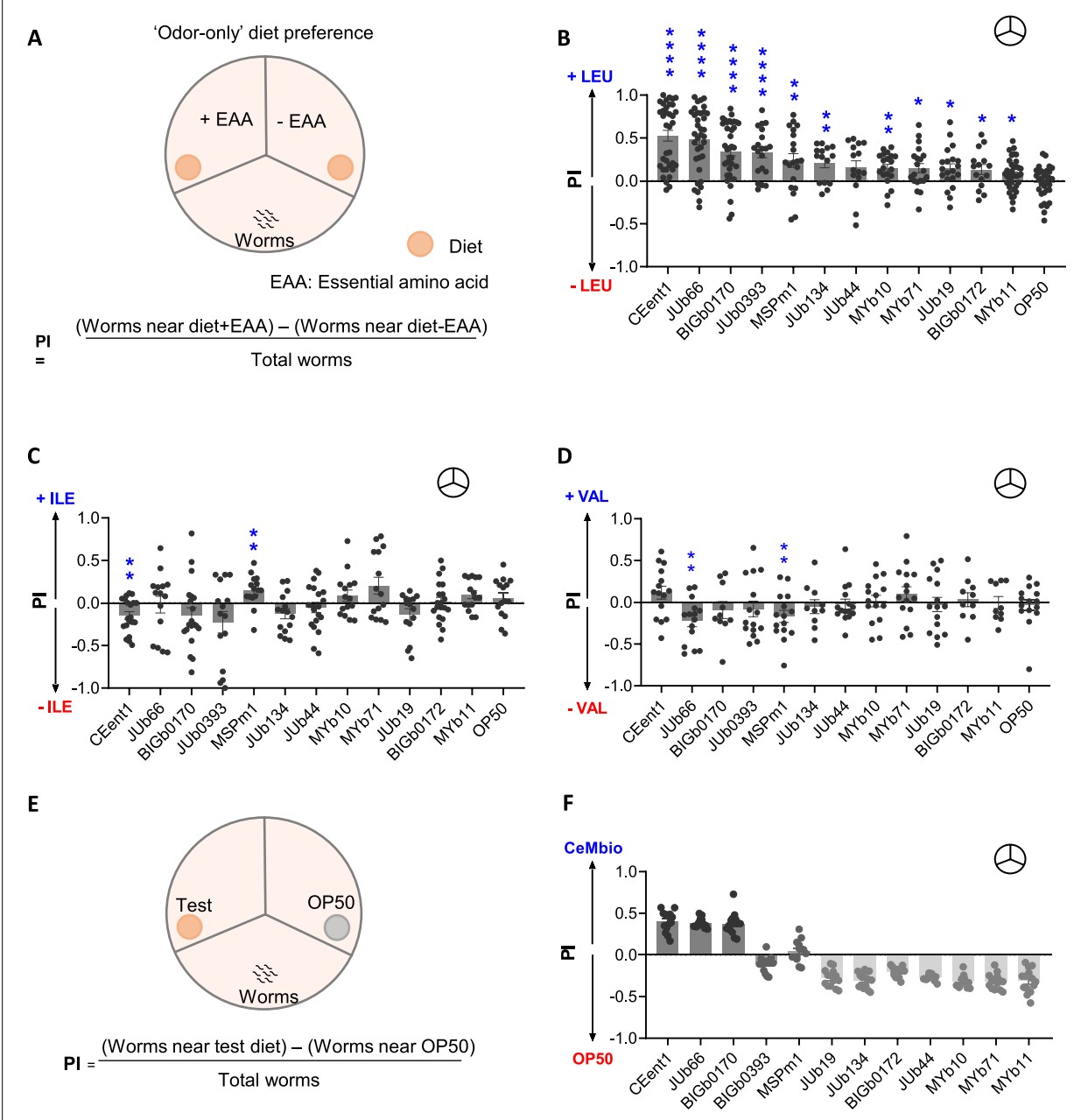

**Figure 1.** *C. elegans* relies on odors to select leucine-supplemented bacteria. (**A**) Schematic representation of 'odor-only' diet preference assay. Preference index (PI) of wild-type (WT) worms in an 'odor-only' diet preference assay (indicated as ⊘) for individual diet supplemented with (**B**) 5 mM leucine (+LEU), (**C**) 5 mM isoleucine (+ILE), and (**D**) 5 mM valine (+VAL). Significant differences are indicated as *p≤0.05, **p≤0.01, and ****p≤0.0001 determined by one-sample *t*-test. (**E**) Schematic representation of 'odor-only' diet preference assay between individual CeMbio bacteria and *E. coli* OP50. (**F**) Preference index (PI) of WT worms in an 'odor-only' diet preference assay between individual CeMbio bacteria and *E. coli* OP50. Also, see *Figure 1—video 1* for diet preference assay. Error bars indicate SEM (n≥15).

The online version of this article includes the following video and figure supplement(s) for figure 1:

**Figure supplement 1.** *C. elegans* does not prefer some essential amino acid (EAA)-supplemented diets.

**Figure 1—video 1.** WIld-type (WT) worms prefer CEent1 over *E. coli* OP50 in a diet preference assay.

https://elifesciences.org/articles/101936/figures#fig1video1

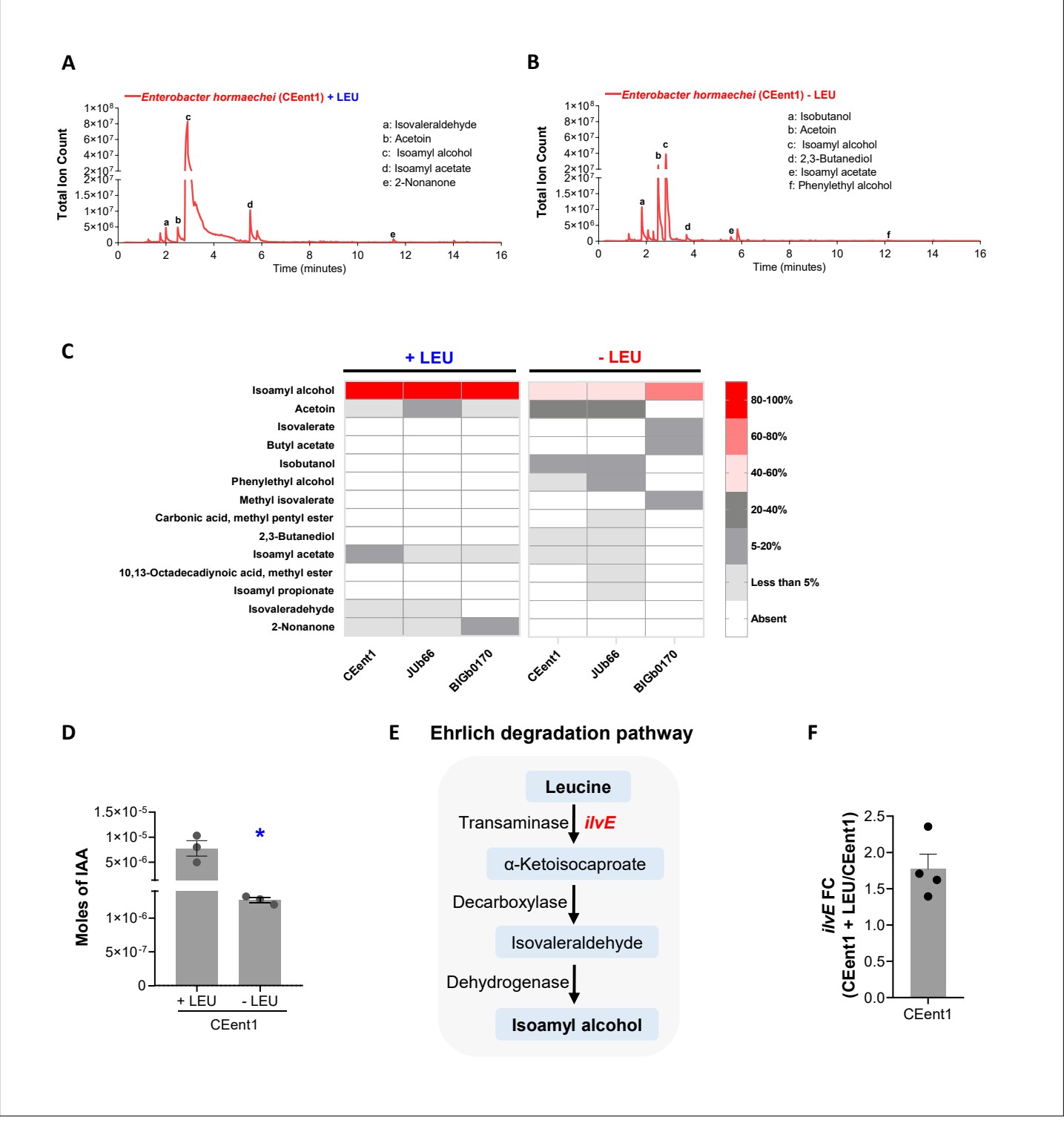

**Figure 2.** Leucine supplementation boosts isoamyl alcohol levels via the Ehrlich degradation pathway. Gas chromatography-mass spectrometry (GC-MS/MS) profile of odors produced by CEent1 under (**A**) leucine-supplemented and (**B**) leucine-unsupplemented conditions. Unmarked peaks represent masses contributed by fiber or media alone (n≥3). (**C**) Heat map representing the relative abundance of various odors in the headspace of CEent1, JUb66, and BIGb0170 with and without leucine supplementation. (**D**) Absolute abundance of isoamyl alcohol (IAA) (in moles) produced by CEent1 under leucine-supplemented and unsupplemented conditions. *$p \leq 0.05$ as determined by a two-tailed unpaired $t$-test. (**E**) Schematic of the Ehrlich degradation pathway. (**F**) Fold change of transcript levels of *ilvE* under leucine-supplemented over unsupplemented conditions for CEent1. Error bars indicate SEM (n≥3).

*Figure 2 continued on next page*

*Figure 2 continued*

The online version of this article includes the following figure supplement(s) for figure 2:

**Figure supplement 1.** Increased levels of isoamyl alcohol upon leucine-enriched influence worms' diet preference.

Microbes can produce short-chain alcohols from carbon sources as well as branched-chain amino acids (*Weisskopf et al., 2021*). Leucine can be catabolized to IAA via the Ehrlich degradation pathway in three enzymatic steps. The first step is catalyzed by IlvE, a transaminase, which converts leucine to α-ketoisocaproate (schematic in *Figure 2E*). The *ilvE* or equivalent transaminase-encoding gene is present in the genome of all CeMbio bacterial strains, including the preferred strains CEent1, JUb66, and BIGb0170 (*Dirksen et al., 2020*). We predicted that leucine supplementation would result in the upregulation of *ilvE* in a substrate-dependent manner, particularly in preferred bacteria. Using qRT-PCR, we found that transcript levels of *ilvE* were two-fold higher in leucine-supplemented CEent1 over unsupplemented (*Figure 2F*). This indicated that the Ehrlich degradation pathway is functional in IAA-producing bacteria and further stimulated in response to leucine. Altogether, these findings show that IAA produced via Ehrlich degradation is a signature for a leucine-supplemented bacteria.

## AWC odor sensory neurons facilitate the diet preference of *C. elegans* for leucine-supplemented diets

Worms use two pairs of odor sensory neurons, AWA and AWC, to sense attractive olfactory cues (*Bargmann, 2006*; *Bargmann et al., 1993*; *Troemel et al., 1997*). To identify the odor sensory neurons that mediate preference for leucine-supplemented diets, we studied the response of *odr-7* worms, which do not have functional AWA neurons, and AWC ablation (-) worms (*Beverly et al., 2011*; *Sengupta et al., 1994*). We found that the preference of worms for leucine-supplemented diets was dramatically lost in AWC(-) worms but was retained in *odr-7* worms (*Figure 3A–C*). This suggested that AWC neurons play a crucial role in sensing odors from leucine-supplemented diets. We also tested the role of these neurons in mediating diet preference between preferred CeMbio bacteria and *E. coli* OP50. We found that AWC(-) worms showed reduced preference for CEent1, JUb66, and BIGb0170 over *E. coli*, while *odr-7* mutants had slightly diminished preference for CEent1 and BIGb0170 over *E. coli* (*Figure 3—figure supplement 1A–C*). This suggests AWC, largely, and AWA odor sensory neurons contribute to the diet preference of worms for specific microbes in their natural habitat (*Figure 3—figure supplement 1D*). The finding that diet preference entails two pairs of neurons suggests that multiple odors constitute a foraging cue enabling worms to find diet bacteria.

To identify the constituents of the foraging signal, we sampled the headspace of each CeMbio bacterium. Each bacterium produced a unique bouquet of odors composed of chemically diverse molecules, including alcohols, aldehydes, ketones, esters, and carboxylic acids (*Supplementary file 2*). To identify foraging cues amongst odors in the headspace of preferred bacteria, we examined the response of WT worms to ten different odors in chemotaxis assays (schematic in *Figure 3—figure supplement 2A*). We found that WT worms were attracted to IAA, acetoin (ACE), isovalerate (ISV), isobutanol (ISB), and phenylethyl alcohol (PEA) in a dose-dependent manner (*Figure 3—figure supplement 2B–F*), of which IAA has been reported to be a strong attractant (*Bargmann and Horvitz, 1991*). Worms showed attraction to butyl acetate (BA) and methyl isovalerate (MIV) only at specific concentrations but did not display a clear dose-response (*Figure 3—figure supplement 2G, H*). Worms showed no response to 2,3-butanediol (BDL), isoamyl acetate (ISA), and indole (IND) (*Figure 3—figure supplement 2I–K*). We would like to note that this is the first report of ACE and MIV as attractants for *C. elegans*. These results show that the headspace of preferred bacteria contains multiple attractive odors for *C. elegans*.

To identify neurons responsible for sensing attractive odors produced by preferred bacteria, we used the *odr-7* and AWC(-) worms in chemotaxis assays for each of the seven attractive odors. We found that AWC(-) worms had severely diminished responses to IAA, ISV, ISB, PEA, BA, and MIV (*Figure 3D and F–J*). ACE, on the other hand, was sensed by AWA neurons as *odr-7* showed reduced response to acetoin (*Figure 3E*). Taken together, our findings revealed that worms sense an extensive repertoire of chemically diverse odors produced by preferred bacteria as attractants, the majority of them using AWC neurons (summary in *Figure 3K*).

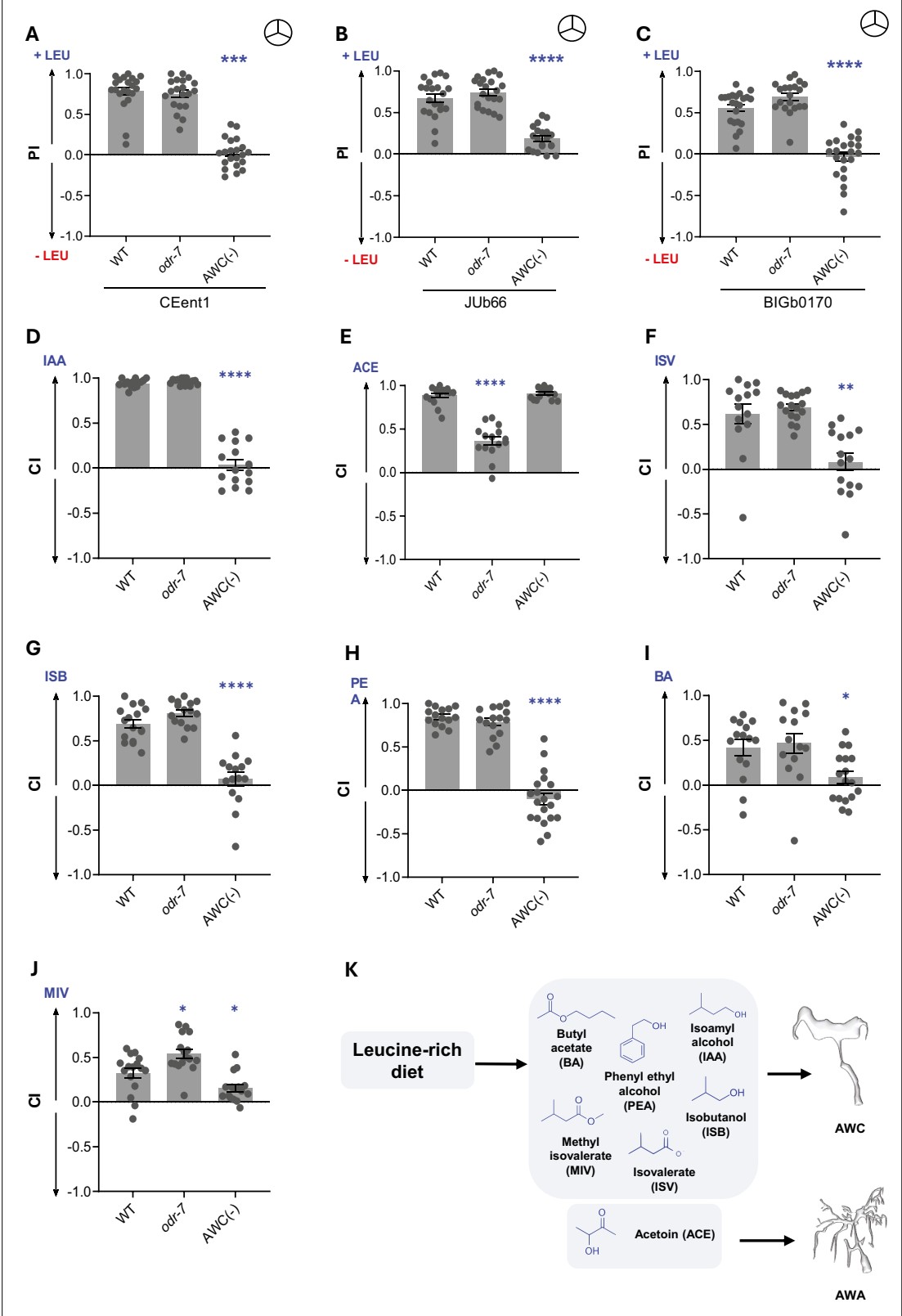

**Figure 3.** Odors sensed through AWC neurons mediate the diet preference of *C. elegans*. Preference index (PI) of wild-type (WT), *odr-7*, and AWC(-) worms for (**A**) CEent1, (**B**) JUb66, and (**C**) BIGb0170 supplemented with leucine over unsupplemented conditions. Symbol ⊘ indicates 'odor-only' preference assays. Chemotaxis index (CI) of WT, *odr-7*, and AWC(-) worms for (**D**) isoamyl alcohol (IAA), (**E**) acetoin (ACE), (**F**) isovalerate (ISV), (**G**) isobutanol (ISB), (**H**) phenylethyl alcohol (PEA), (**I**) butyl acetate (BA), and (**J**) methyl isovalerate (MIV). (**K**) Summary schematic representing the role of

*Figure 3 continued on next page*

*Figure 3 continued*

AWA and AWC odor sensory neurons in 'odor-only' diet preference for leucine-supplemented diets and chemotaxis assays. Significant differences are indicated as *p≤0.05, **p≤0.01, ***p≤0.001, and ****p≤0.0001 determined by one-way ANOVA followed by post hoc Dunnett's multiple comparison test. Error bars indicate SEM (n=15).

The online version of this article includes the following figure supplement(s) for figure 3:

**Figure supplement 1.** AWA and AWC neurons facilitate the diet preference of *C. elegans*.

**Figure supplement 2.** Dose-dependent chemotaxis response of *C. elegans* to individual odors produced by preferred bacteria.

## Isoamyl alcohol drives foraging behavior in *C. elegans*

To identify the most relevant foraging signal for *C. elegans*, we took advantage of the well-established odor adaptation assay using each of the attractive odors in its diet (schematic in *Figure 4A*). Prolonged exposure of *C. elegans* to an odorant is known to result in the loss or reduction of the ability of worms to sense the same odorant in a subsequent exposure (*Colbert and Bargmann, 1995*). We performed a modified adaptation assay where worms were exposed to a bouquet of odors from a bacterial lawn (- LEU), followed by testing their chemotaxis response to individual attractive odors. Worms adapted to odors from CEent1 lawns had dramatically reduced responses to IAA, isobutanol, and phenylethyl alcohol compared to naïve worms (*Figure 4B, E and F*). Worms adapted to odors from JUb66 and BIGb0170 lawns also had dramatically reduced response to IAA (*Figure 4B*). A adaptation to bacterial odors did not influence worms' response to ACE, ISV, BA, and MIV (*Figure 4C, D, G and H*). These findings suggested that the odor bouquets of preferred bacteria have levels of IAA high enough to modulate chemoperception in *C. elegans*.

If IAA was the most relevant foraging cue, we predicted that prior adaptation to IAA would diminish the diet preference of *C. elegans*. To test this, we adapted worms to IAA odor and tested their diet preference (schematic in *Figure 4—figure supplement 1A*). IAA-adapted worms showed a reduced chemotaxis response to IAA, indicating that the adaptation regimen is robust (*Figure 4—figure supplement 1B*). We found that IAA-adapted worms displayed a reduced preference for CEent1, JUb66, or BIGb0170 over *E. coli* OP50 compared to naïve worms (*Figure 4—figure supplement 1C–E*). These results indicated that IAA was the predominant olfactory cue that determines the diet preference of *C. elegans*.

## IAA is an ecologically relevant odor for *C. elegans*

If an odor is relevant as a foraging cue in nature, the ability to sense it must be under positive selection in wild *C. elegans* populations. We anticipated that most wild isolates of *C. elegans* will have a robust response to IAA but not to other attractants reported in this study. To test this hypothesis, we studied the response to IAA in nine phylogenetically distinct wild isolates of *C. elegans* collected from geographically distinct regions (*Figure 5A* and *Supplementary file 3*; *Andersen et al., 2012*; *Cook et al., 2017*). Additionally, we tested their chemotaxis response to diacetyl, a known food cue, and two other attractants (PEA and ACE) identified in this study. We found that all the strains displayed robust responses to IAA comparable to WT worms (*Figure 5B*). The chemotaxis response of these strains to diacetyl was also quite robust (*Figure 5C*). However, the chemotaxis response to PEA and acetoin were variable within and across wild isolates, suggesting they may not be crucial for foraging (*Figure 5D and E*). The robust response of wild isolates to IAA supports the notion that the ability of *C. elegans* to sense IAA has been selected during evolution. These findings suggest that IAA is an ecologically relevant foraging cue for *C. elegans*.

## SNIF-1 senses isoamyl alcohol and facilitates foraging in *C. elegans*

*C. elegans* genome encodes for ~1300 putative GPCRs, many of which are predicted to sense soluble and olfactory cues (*Thomas and Robertson, 2008*). Since the preference for IAA and leucine-supplemented diets is mediated by AWC odor sensory neurons (*Figure 3A–D*), we hypothesized that the receptor for IAA is a GPCR expressed in AWC neurons. Indeed, calcium response to IAA has been reported in AWC neurons (*Zaslaver et al., 2015*; *Lin et al., 2023*; *Bargmann and Horvitz, 1991*). Using single-cell RNAseq data for *C. elegans* neurons, available at CeNGEN (*Hammarlund et al., 2018*), we listed 18 putative GPCRs expressed in AWC[on] and AWC[off] for further analysis (*Figure 6A*). We tested response to IAA in 15 strains harboring mutants in one or more of these GPCRs (see

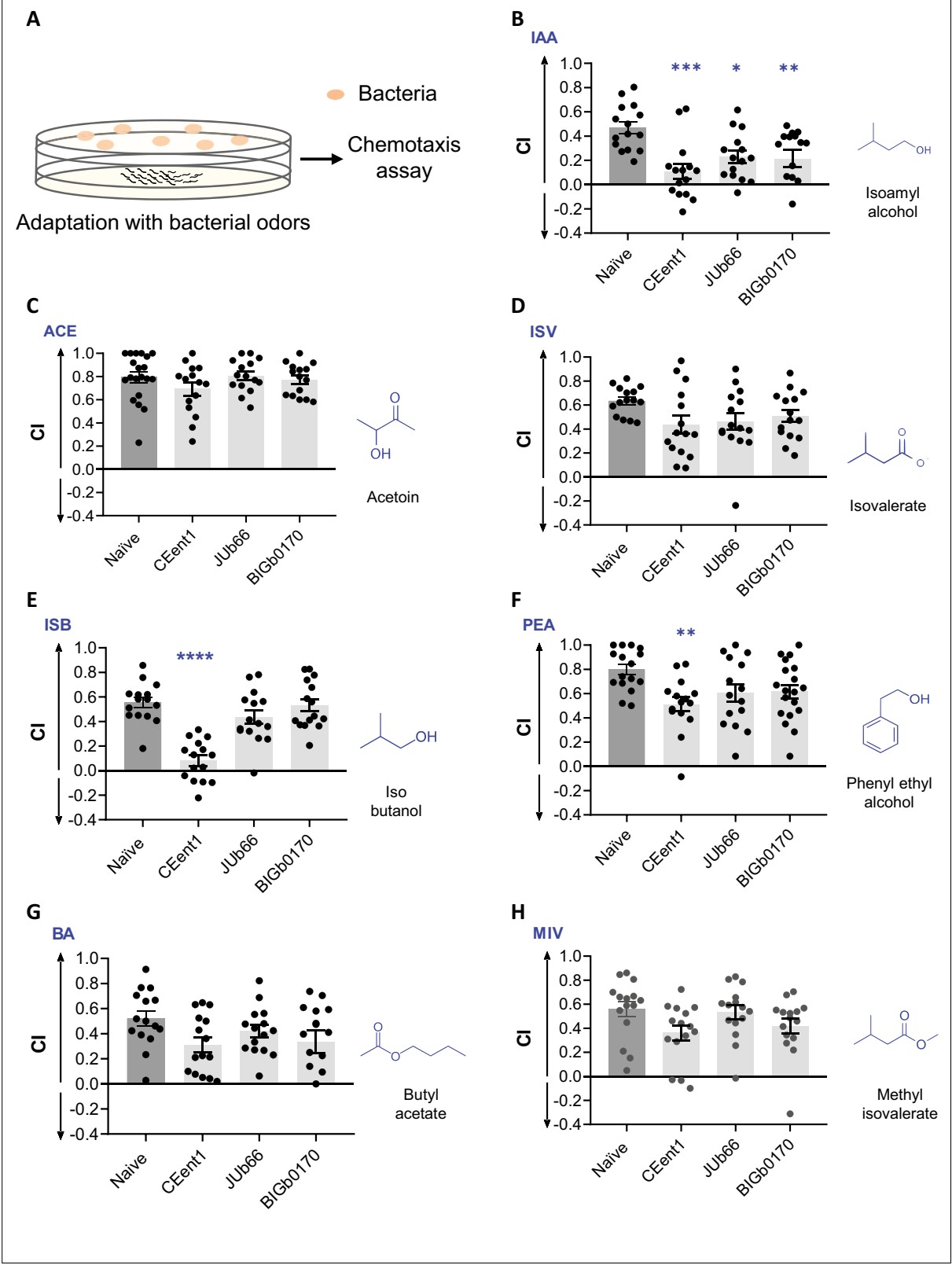

**Figure 4.** Isoamyl alcohol regulates the diet preference of worms. (**A**) Schematic representation of the adaptation regimen followed by chemotaxis assays. Chemotaxis index (CI) for naïve worms and worms adapted with CEent1, JUb66, or BIGb0170 odors to (**B**) isoamyl alcohol (IAA), (**C**) acetoin (ACE), (**D**) isovalerate (ISV), (**E**) isobutanol (ISB), (**F**) phenylethyl alcohol (PEA), (**G**) butyl acetate (BA), and (**H**) methyl isovalerate (MIV). Dark gray bars

*Figure 4 continued on next page*

*Figure 4 continued*

indicate naïve worms, and light gray bars indicate worms adapted to bacterial odors. Significant differences are indicated as *$p \leq 0.05$, **$p \leq 0.01$, ***$p \leq 0.001$, and ****$p \leq 0.0001$ determined by one-way ANOVA followed by post hoc Dunnett's multiple comparison test. Error bars indicate SEM (n=15).

The online version of this article includes the following figure supplement(s) for figure 4:

**Figure supplement 1.** *C. elegans* utilizes isoamyl alcohol to make dietary preferences.

---

*Supplementary file 3*; *Pu et al., 2023b*). One of these strains, CHS1169, had a significantly reduced response to IAA compared to WT worms (*Figure 6A*). Of the six GPCR-encoding genes edited in the CHS1169 strain, *srd-12* had the highest expression in AWC^off odor sensory neuron (*Hammarlund et al., 2018*). Using a CRISPR/Cas9-based approach, we generated two *srd-12* deletion lines, VSL2401 and VSL2402, carrying mutations in the *srd-12* gene (see methods and *Supplementary file 3*). We recorded the chemotaxis response of WT and mutant worms to IAA (schematic in *Figure 6B*) and found they were indeed defective in their response to IAA (*Figure 6C–F*). By analyzing the locomotion of WT, VSL2401, and VSL2402 worms in the chemotaxis arena, we determined quantitative differences in their response to IAA. We observed that VSL2401 and VSL2402 strains had speeds comparable to WT worms and traveled distances similar to WT worms, suggesting that *srd-12* mutants do not have locomotory defects (*Figure 6—figure supplement 1A, B*). We found that a mutation in the *srd-12* gene resulted in significantly reduced chemotaxis response to IAA (*Figure 6F*). However, *srd-12* mutants had a comparable response to WT worms for ACE and PEA (*Figure 4—figure supplement 1C, D*). This result suggested that mutation in *srd-12* causes a defect in sensing IAA alone. We rename the *srd-12* gene and encoded protein as *snif-1* and sniffing receptor SNIF-1, respectively. As expected, *snif-1* mutant strains had diminished preference for CEent1 over *E. coli* OP50 in the diet preference assay, consistent with the idea that IAA is the most relevant foraging signal (*Figure 6G*).

To confirm that IAA receptor is indeed functional in AWC neuron, we generated transgenic worms carrying extrachromosomal array expressing *snif-1* cDNA under its native promoter (*snif-1p::snif-1*) as well as under an AWC-specific promoter (AWCp::*snif-1*). We found that transgenic lines expressing SNIF-1 under its native promoter or AWC-specific promoter completely rescued the chemotaxis response of the *snif-1* mutant worms to IAA (*Figure 6H*). To further confirm the function of SNIF-1 as an IAA receptor, we used a misexpression strategy. While AWA and AWC neurons induce attraction in worms, AWB neurons elicit avoidance response (*Troemel et al., 1997*; *Bargmann, 2006*; *Bargmann et al., 1993*; *Troemel et al., 1997*). We reasoned that the misexpression of SNIF-1 in AWB should cause aversion to the odor. We generated transgenic worms carrying extrachromosomal arrays expressing *snif-1* cDNA under a bona fide AWB-specific promoter (AWBp::*snif-1*). As expected, the misexpression of *snif-1* in AWB neurons elicited repulsion to IAA in worms (*Figure 6H*). To confirm the expression of *snif-1* in AWC neurons, we generated a transgenic line of worms expressing GFP under the *snif-1* promoter and mCherry under the *odr-1* promoter (to mark AWC neurons). We found that *snif-1* is expressed faintly in many neurons, with strong expression in one of the two AWC neurons marked by *odr-1p::mCherry* (*Figure 6I*). Taken together, our findings show that *C. elegans* senses foraging cue, IAA, using SNIF-1 GPCR in one of the AWC neurons.

Based on our study, we propose a model describing an odor-dependent mechanism for sensing EAA-enriched bacteria in *C. elegans*. IAA is an olfactory cue for leucine-supplemented diet. AWC olfactory neurons of *C. elegans* allow worms to choose IAA-producing bacteria. Bacteria preferred by worms have the metabolic capacity to produce IAA from leucine via the Ehrlich degradation pathway. SNIF-1, a GPCR in AWC neurons, mediates foraging for bacteria in *C. elegans* by facilitating chemoperception of IAA (*Figure 7*).

## Discussion

Our study provides evidence for an odor-dependent mechanism for foraging in *C. elegans*. Using CeMbio, a model microbiome for *C. elegans*, we show that worms prefer leucine-supplemented bacteria in an odor-dependent manner. Leucine-supplemented diets produce significantly higher levels of IAA odor, making up to 90% of their headspace. These bacteria produce IAA from leucine using the Ehrlich degradation pathway. Preference for leucine-supplemented bacteria solely depends on the AWC odor sensory neurons of *C. elegans*. Although the preferred bacteria produce several

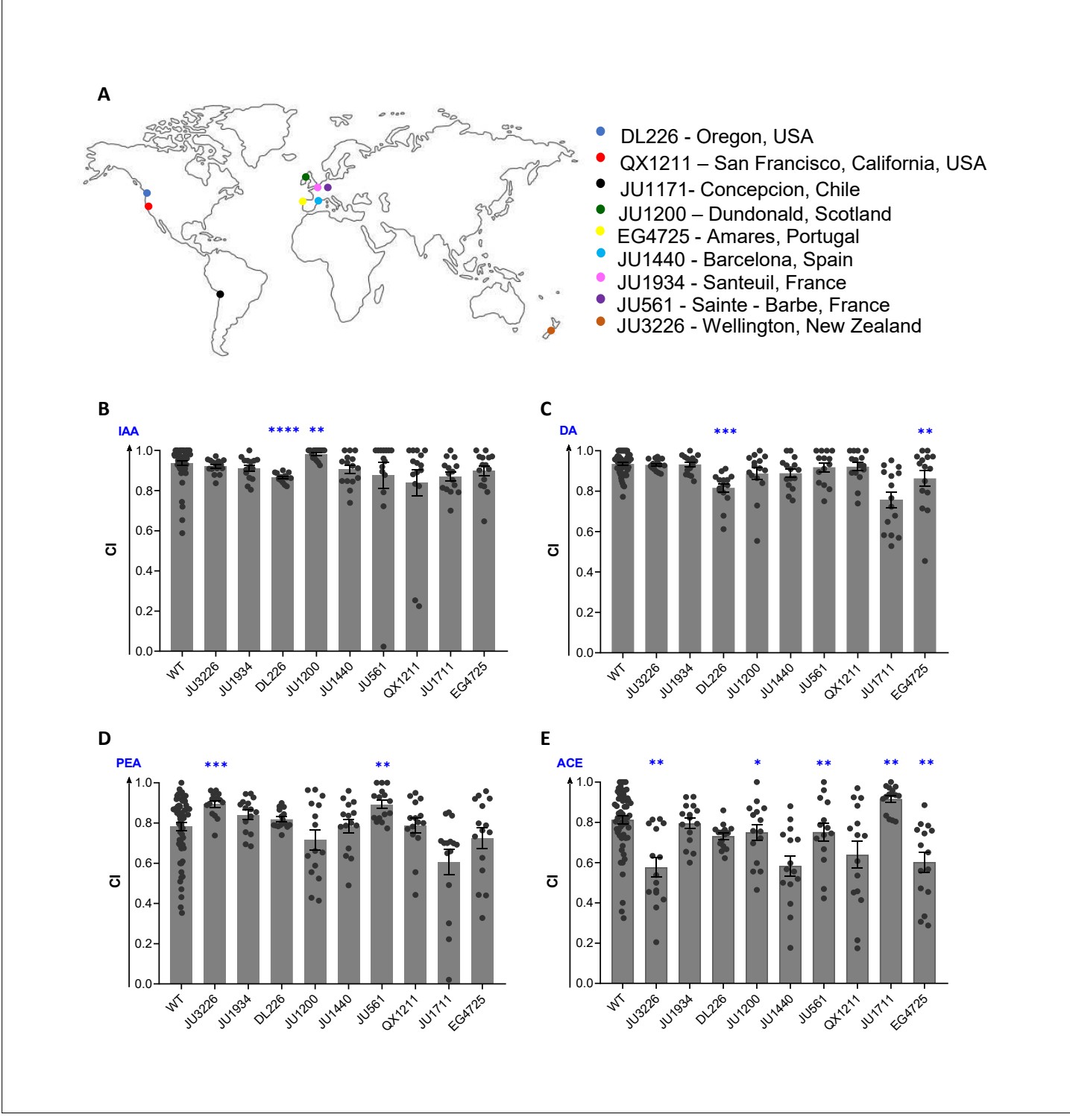

**Figure 5.** Robust chemoperception of isoamyl alcohol in wild isolates of *C. elegans*. (**A**) World map representing the distinct geographical locations (source) of wild isolates of *C. elegans* used in this study. Chemotaxis index (CI) of wild-type (WT) and wild isolates of worms for (**B**) isoamyl alcohol (IAA), (**C**) diacetyl (DA), (**D**) phenylethyl alcohol (PEA), and (**E**) acetoin (ACE). Significant differences are indicated as *$p \le 0.05$, **$p \le 0.01$, ***$p \le 0.001$, and ****$p \le 0.0001$ determined by one-way ANOVA followed by post hoc Dunnett's multiple comparison test. Error bars indicate SEM (n=15).

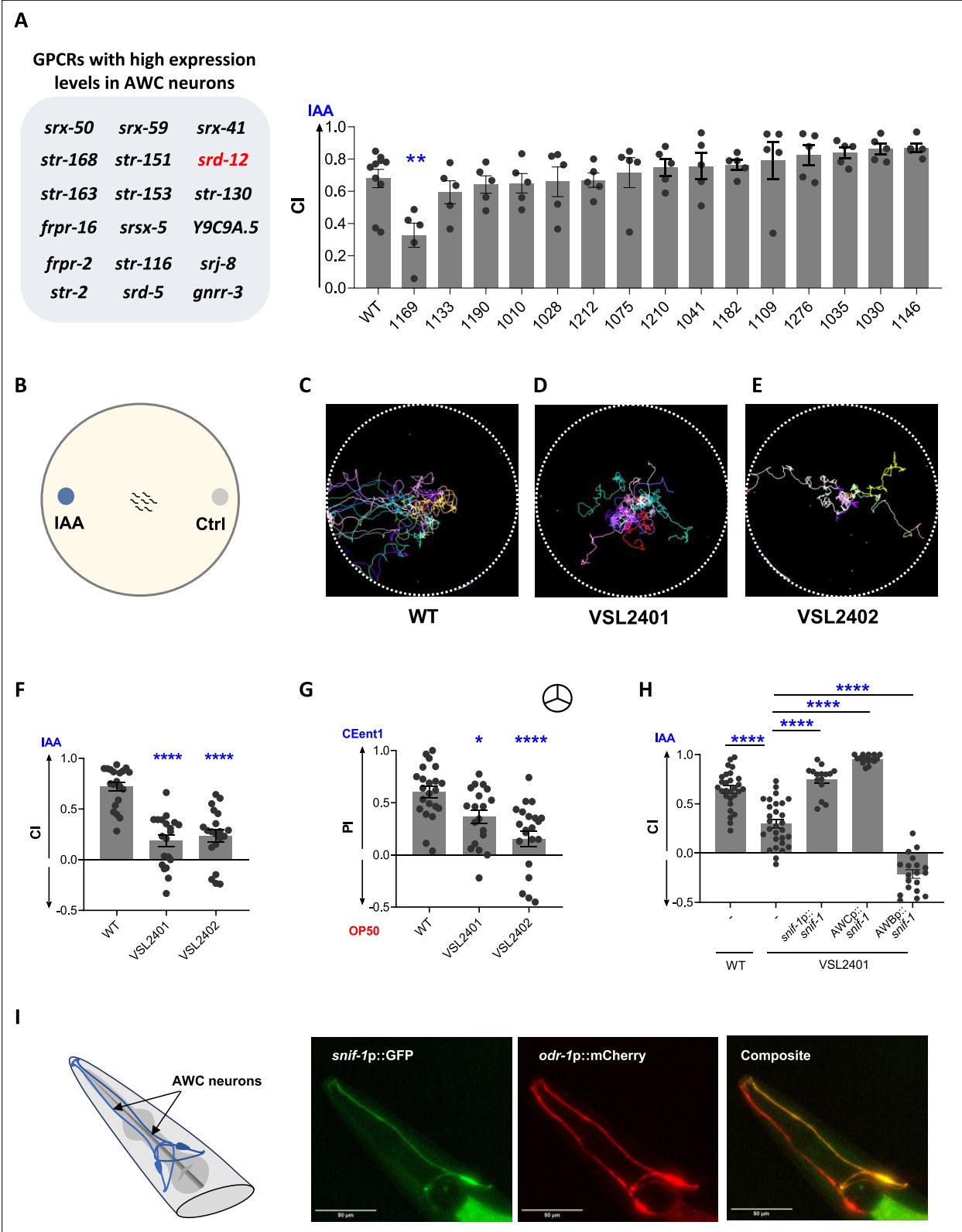

**Figure 6.** SNIF-1 G-protein coupled receptor (GPCR) mediates isoamyl alcohol sensing and diet preference in *C. elegans*. (**A**) List of GPCRs that are highly expressed in AWC neurons. Chemotaxis index (CI) of wild-type (WT) and GPCR-edited strains to 1:1000 isoamyl alcohol (IAA) (n≥3). All GPCR mutants of *C. elegans* used for screening have the CHS designation along with the codes mentioned in the panels. Also, refer to **Supplementary file 3** for strain information. (**B**) Schematic for chemotaxis assay plate used for movement track analysis. Movement tracks of five animals in a chemotaxis arena

*Figure 6 continued on next page*

*Figure 6 continued*

with 1:1000 IAA for 15 min, for (**C**) WT, and *snif-1* mutants- (**D**) VSL2401, and (**E**) VSL2402. Also, refer to *Figure 6—video 1*. Individual color represents the track for a single worm. (**F**) Chemotaxis index (CI) in response to IAA (1:1000 dilution) for WT, VSL2401, and VSL2402 worms. (**G**) Preference index (PI) of WT, VSL2401, and VSL2402 worms for a preferred diet, CEent1, over *E. coli* OP50. ⊘ symbol indicates 'odor-only' preference assays. (**H**) Chemotaxis index (CI) in response to IAA (1:1000 dilution) for WT, VSL2401, VSL2401 [*snif-1*p:: *snif-1*], VSL2401 [AWCp::*snif-1*] and VSL2401 [AWBp::*snif-1*]. Significant differences are indicated as *p≤0.05, **p≤0.01, and ****p≤0.0001 determined by one-way ANOVA followed by post hoc Dunnett's multiple comparison test. Error bars indicate SEM (n≥15). (**I**) Schematic showing the soma and processes of AWC neurons in the amphid region. Representative images of the reporter line expressing GFP under *snif-1* promoter, and mCherry under *odr-1* promoter (to mark AWC neurons) (n≥30).

The online version of this article includes the following video and figure supplement(s) for figure 6:

**Figure supplement 1.** Differential effect of SNIF-1 on locomotion of *C. elegans*.

**Figure 6—video 1.** Chemotaxis response of wild-type (WT), VSL2401, and VSL2402 worms to isoamyl alcohol (IAA).

https://elifesciences.org/articles/101936/figures#fig6video1

AWC-sensed odors, IAA exclusively influences the diet preference of worms. Phylogenetically distinct wild isolates of *C. elegans*, representing nine geographically distinct locations on Earth, respond robustly to IAA underscoring its relevance as a foraging cue in nature. *C. elegans* utilizes SNIF-1, a GPCR primarily expressed in AWC neurons, to sense IAA and forage for the preferred bacteria.

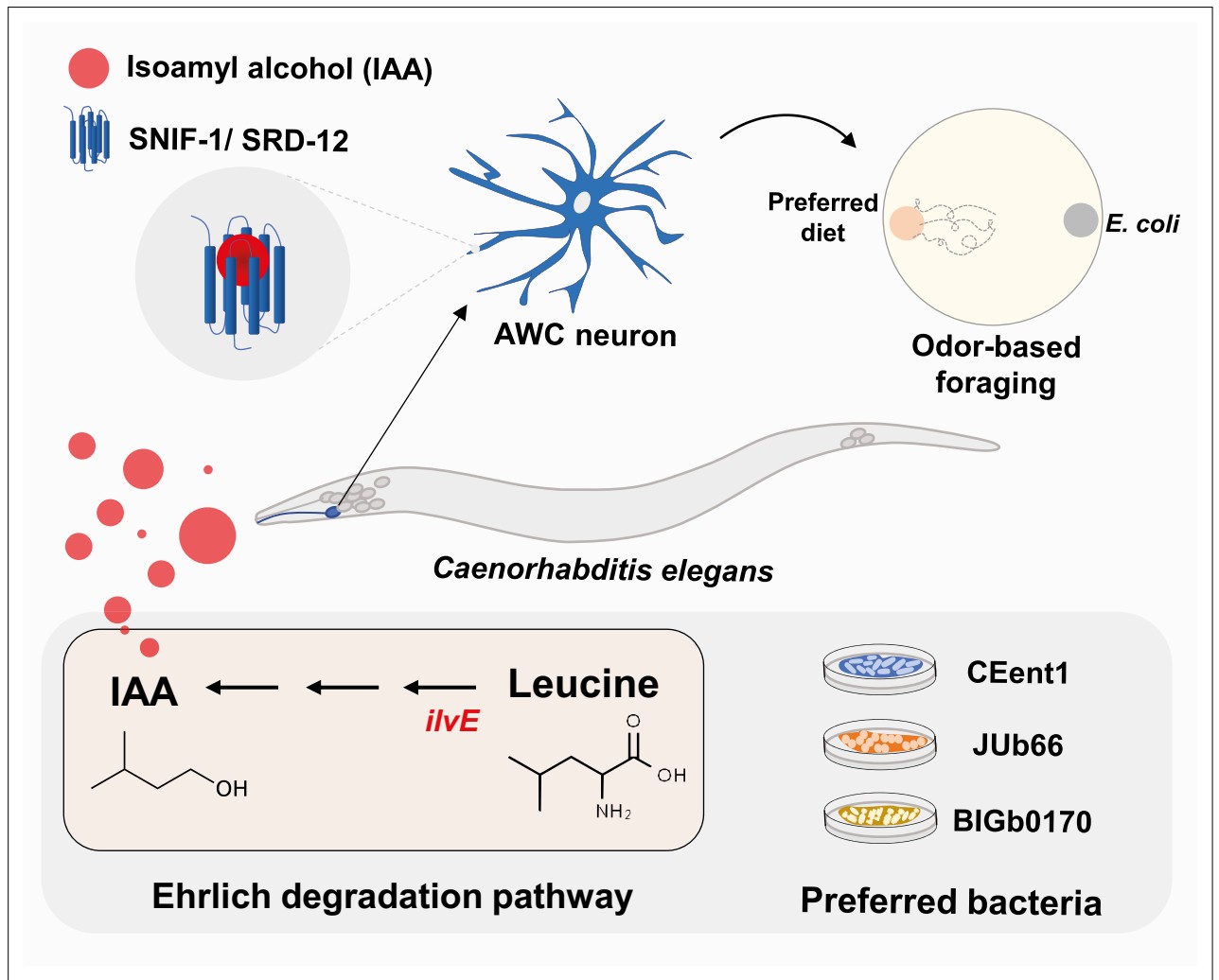

**Figure 7.** Model depicting odor-based foraging strategy used by *C. elegans*. *C. elegans* prefers leucine-supplemented diets in an odor-dependent manner. Preferred bacteria catabolize leucine, an essential amino acid (EAA), to produce isoamyl alcohol (IAA) via the Ehrlich degradation pathway. SRD-12/SNIF-1 G-protein coupled receptor (GPCR) expressed in AWC odor sensory neurons mediates the foraging behavior of *C. elegans* by sensing IAA.

Thus, IAA-SNIF-1 represents a ligand-receptor module used by *C. elegans* to forage for bacteria in its natural environment.

Is olfaction a major contributor to foraging behavior in animals in general? Artificial flowers, containing nectars with high amino acid content, are better at attracting butterflies (*Alm et al., 1990*). Honeybees prefer to consume artificial nectar rich in proline (*Bertazzini et al., 2010*). Hummingbirds, moths, and ants are attracted to flowers emitting benzyl acetone, an odor synthesized from an EAA, phenylalanine (*Haverkamp et al., 2016*; *Yon et al., 2016*; *Bhattacharya and Baldwin, 2012*; *Kessler and Baldwin, 2007*). For odors to serve as an attractant, the said odor must serve as a proxy or code for a needed primary metabolite, abundant in carbon, nitrogen, phosphorus, or micronutrients. In such a scenario, it is easy to envisage that each attractive cue might be a code for a specific essential nutrient.

A foraging signal or cue is any sensory stimulus which helps an animal find and evaluate food. While 'signals' are stimuli that have specifically evolved for the receiver, 'cues' are rather passive traits that may not be directed towards the receiver (*Lehmann et al., 2014*). For instance, *Brassica rapa* plants that are pollinated by bumble bees exhibit evolved characteristics, such as increased height, double the amount of odors, and larger UV-reflecting petals, all of which serve as foraging signals for bumble bees (*Gervasi and Schiestl, 2017*). In contrast, hawkmoths utilize transient humidity changes in the headspace of *Oenothera cespitosa* flowers as a foraging cue for identifying nectar-rich flowers (*von Arx et al., 2012*). Is isoamyl alcohol a foraging cue for essential nutrient for *C. elegans* and other foraging animals? IAA was one of 61 attractive odors reported to be sensed by *C. elegans* several decades ago (*Bargmann et al., 1993*), but its ecological relevance was unclear to date. A recent study also showed that worms specifically prefer IAA-producing bacteria (*Chai et al., 2024*; *Worthy et al., 2018b*). Our study suggests that IAA is the primary cue that drives foraging in favor of leucine-supplemented diets in *C. elegans*. Does IAA reflect the metabolic state of bacteria producing it? IAA is indeed derived from a primary metabolite leucine, an essential branched-chain amino acid for animals (*Hazelwood et al., 2008*). In the Ehrlich degradation pathway, leucine is first converted to alpha-keto acid using a transaminase (IlvE), which undergoes decarboxylation and reduction in the following steps. While *ilvE* or similar genes are present in all CeMbio bacteria, not all of them produce IAA (*Supplementary file 2*). It is possible that the remaining bacteria lack enzymes for decarboxylation and reduction of alpha-keto acid. When bacteria were provided with additional leucine, the absolute abundance of IAA increased significantly in three different bacteria examined (*Figure 2C* and *Figure 2—figure supplement 1F, G*). CEent1, one of the preferred bacteria, also showed a substrate-dependent upregulation of *ilvE* transcript (*Figure 2F*). The arrival of feeding insects to ripening fruits, such as bananas coincides with the production of IAA and isoamyl acetate in the fruit via the Ehrlich degradation pathway (*Dou et al., 2022*). The presence of the Ehrlich degradation pathway in specific microbes and many insect-attracting plants underscores the importance of odors in mediating inter-kingdom interactions. IAA, in combination with other odors, has been reported to attract *Drosophila melanogaster* and help in establishing a fly-fruit-yeast relationship (*Becher et al., 2012*; *Dötterl and Gershenzon, 2023*; *He et al., 2022*; *Liu et al., 2022*; *Zjacic and Scholz, 2022*; *Haupt et al., 2010*; *Pham-Delegue et al., 1993*). Why do animals need BCAA? Not only is leucine a major component of *C. elegans* proteome, intracellular accumulation of BCAA, including leucine prolongs the lifespan of *C. elegans*. *Edwards et al., 2015* have reported a 15% increase in the lifespan of worms upon 1 mM leucine supplementation (*Edwards et al., 2015*). *Wang et al., 2018* reported that along with lifespan extension, leucine supplementation makes worms more resistant to heat, paraquat, and UV stress (*Wang et al., 2018*). The lack of response to leucine (*Figure 1—figure supplement 1H* in this study) in the presence of IAA perception ability, via SNIF-1, suggests that worms rely on IAA sensing to identify bacteria with leucine. Additional natural microbiotas of *C. elegans* (not part of CeMbio) which produce IAA have also been reported to attract worms (*Worthy et al., 2018a*; *Worthy et al., 2018b*). Worms also prefer odors derived from other BCAA, such as isobutanol derived from valine via Ehrlich degradation (*Figure 4E*; *Bizzio et al., 2021*; *Maoz et al., 2022*; *Yan et al., 2022*). Phenylethyl alcohol (PEA in this study) can also be produced by Ehrlich degradation of phenylalanine, another essential amino acid that animals require for their proteins as well as for the synthesis of neurotransmitter dopamine. All these reports support the idea that animals use olfactory cues to identify diets rich in essential amino acids.

Animals use GPCRs to sense soluble and olfactory cues present in their environment. Rats utilize taste receptors T1R1/T1R3 for sensing alanine (*Taylor-Burds et al., 2004*). *C. elegans* can also sense lysine and histidine (*Ward, 1973*) but not leucine (this study). Several studies have implicated olfactory GPCRs in regulating animal behavior. In termites, olfactory co-receptor ORCO influences foraging by enabling pheromone sensing (*Xu et al., 2024*). Or83b, an odorant receptor in *D. melanogaster,* regulates lifespan (*Libert et al., 2007*). Six odor sensory neurons of *C. elegans* express hundreds of GPCRs, most of them remain orphan (*Hammarlund et al., 2018*). ODR-10 expressed in AWA odor sensory neurons regulates foraging behavior in *C. elegans* for lactic acid bacteria which produce diacetyl odor (*Choi et al., 2016*). AWC odor sensory neurons express as many as 181 GPCRs, of which 18 GPCRs display strong expression (*Hammarlund et al., 2018*). These receptors likely mediate the foraging behavior of worms. One of these, STR-2, senses 2-heptanone odor produced by *Bacillus nematocidal* (*Zhang et al., 2016*). STR-2 also regulates lipid accumulation and lifespan in a temperature-dependent manner (*Dixit et al., 2020*). The ligand(s) for the remaining 17 receptors were unknown. The discovery of one of these, SNIF-1/SRD-12, as a receptor for sensing IAA (this study) suggests that the ligand for other receptors may also be found in the headspace of the *C. elegans* microbiota. Our study provides a framework for the identification of ligands for orphan GPCRs by searching for an ecologically relevant repertoire, the microbiota of an animal.

## Methods

**Key resources table**

| Reagent type (species) or resource | Designation | Source or reference | Identifiers | Additional information |
|---|---|---|---|---|
| Gene (*Caenorhabditis elegans*) | *srd-12* | Wormbase | WormBase ID: WBGene00005090 | Renamed in this article as *snif-1* |
| Strain, strain background (*Caenorhabditis elegans*) | N2 | CGC | CGC reference 257 | |
| Sequence-based reagent | *snif-1* promoter for position 1_F | This paper | Cloning PCR primers | GGGGACAACTTTGTATAGAAAAGTTGtctagtgtaagagtgaaacttg |
| Sequence-based reagent | *snif-1* promoter for position 1_R | This paper | Cloning PCR primers | GGGGACTGCTTTTTTGTACAAACTTGgactttgatttctgaaaaagaaaa |
| Sequence-based reagent | *snif-1* for position 2_F | This paper | Cloning PCR primers | GGGGACAAGTTTGTACAAAAAAGCAGGCTATGTTTGATTTTGCTGTTCTCTTCT |
| Sequence-based reagent | *snif-1* for position 2_R | This paper | Cloning PCR primers | GGGGACCACTTTGTACAAGAAAGCTGGGTctcaTTATCCAATAATATGAATAC |
| Sequence-based reagent | *snif-1*_F | This paper | genotyping PCR primers for VSL2401 and VSL2402 | TCAGAAATCAAAGTCATGTTTGATTTTGCTG |
| Sequence-based reagent | *snif-1*_R | This paper | genotyping PCR primers for VSL2401 and VSL2402 | CCTGAGACCAGATGAAGGAGATCCATT |
| Sequence-based reagent | *ilvE*_F | This paper | qRT-PCR primers | TGTGATCATTGCTGCGTTCC |
| Sequence-based reagent | *ilvE*_R | This paper | qRT-PCR primers | CCAGCAGTGAGGAGAGGTAG |
| Sequence-based reagent | *rpoB*_F | This paper | qRT-PCR primers | ACCTGACCAAATACACCCGT |

*Continued on next page*

*Continued*

| Reagent type (species) or resource | Designation | Source or reference | Identifiers | Additional information |
|---|---|---|---|---|
| Sequence-based reagent | *rpoB*_R | This paper | qRT-PCR primers | GATCTTCCTGAACCACACGC |
| Commercial assay or kit | iScript cDNA synthesis kit | Bio-Rad | 170–8891 | |
| Commercial assay or kit | RNeasy Mini kit | Qiagen | 74104 | |
| Chemical compound, drug | CBR-5884 | Sigma Aldrich | SML1656 | |
| Chemical compound, drug | 2,3-Butanediol | Sigma | 237639–1 G | |
| Chemical compound, drug | Acetoin | Sigma | 40,127 U | |
| Chemical compound, drug | Agar | HiMedia | GRM026 | |
| Chemical compound, drug | Butyl acetate | Sigma | 287725–100 ML | |
| Chemical compound, drug | CaCl2 | SRL | 10035-04-8 | |
| Chemical compound, drug | Chloroform | VWR Chemicals | BDH83626.400 | |
| Chemical compound, drug | Diacetyl | Sigma | B85307 | |
| Chemical compound, drug | Difco Luria-Bertani | BD Difco | 11778902 | |
| Chemical compound, drug | DNase I | NEB | M0303S | |
| Chemical compound, drug | Ethanol | Merck Millipore | 100983 | |
| Chemical compound, drug | Indole | Sigma | I3408-100G | |
| Chemical compound, drug | Isoamyl acetate | Sigma | W205532-SAMPLE-K | |
| Chemical compound, drug | Isoamyl alcohol | Sigma | W205710-SAMPLE-K | |
| Chemical compound, drug | Isobutanol | Sigma | 294829–100 ML | |
| Chemical compound, drug | Isovalerate | TCI | 503-74-2 | |
| Chemical compound, drug | K2HPO4 | SRL | 2139900 | |
| Chemical compound, drug | KH2PO4 | SRL | 7778-77-0 | |
| Chemical compound, drug | Methyl isovalerate | TCI | 556-24-1 | |
| Chemical compound, drug | MgSO4 | SRL | 10034-99-8 | |
| Chemical compound, drug | NaCl | SRL | 7647-14-5 | |

*Continued on next page*

*Continued*

| Reagent type (species) or resource | Designation | Source or reference | Identifiers | Additional information |
|---|---|---|---|---|
| Chemical compound, drug | Phenylethyl alcohol | Sigma | 77861–250 ML | |
| Chemical compound, drug | RNAprotect Bacteria Reagent | Qiagen | 76506 | |
| Chemical compound, drug | Sodium azide | Sigma | 71290–10 G | |
| Chemical compound, drug | SYBR Green detection mix | Bio-Rad | 1725124 | |
| Software, algorithm | MATLAB | | | Release R2022b |

## Experimental model and subject details

### Strains and growth media

*C. elegans* strains used in this study are listed in *Supplementary file 3*. All strains were maintained as hermaphrodites at 20 °C on nematode growth media (NGM) plates seeded with *E. coli* OP50, as originally described (*Brenner, 1974*). All the CeMbio strains used in this study (see *Supplementary file 1*) were grown on Difco Luria-Bertani (LB) media and maintained at 25 °C, as described by *Dirksen et al., 2020*. A list of all the reagents used in this study is provided in the Key resource table.

### CRISPR approach for gene editing

Homology-directed integration of the ssDNA oligo was done in the *srd-12 (snif-1)* gene using a CRISPR/Cas9-based approach as described previously (*Pu et al., 2023a*). ssDNA oligo was designed to contain two stop codons along with a restriction site between two 35 bp long homology arms flanking the PAM sites of the targeted gene. ssDNA oligo was incubated with optimized RNP complexes of Cas9, crRNA, and tracrRNA. The mix was injected into the gonad of *C. elegans* along with *rol-6* [pRF4::rol-6(su1006)] marker plasmid. The F1 progenies were segregated and genotyped using PCR, followed by restriction digestion for the introduction of mutation. *srd-12* CRISPR mutants were back-crossed three times with N2 Bristol WT strain to generate VSL2401 and VSL2402.

### Molecular biology

The expression constructs used in this study were generated using the Multisite Gateway System (ThermoFisher Scientific). The promoters were amplified from genomic DNA and cloned in vector P4P1. For expressing *srd-12 (snif-1)* under its native promoter, 2066 bp upstream region to CDS of *srd-12 (snif-1)* was used. The *srd-12 (snif-1)* encoding region was amplified using cDNA isolated from wild-type worms and cloned in pDONR221. LR reactions were performed to assemble the final construct. All constructs were verified by sequencing.

For generating transgenic lines, adult worms with a single row of eggs were injected with a mix consisting of cDNA-expressing constructs and co-marker at 50 ng/µl and 180 ng/µl, respectively. Coelomocyte marker (*unc-122*p::GFP) was used as a co-injection marker. Three to five independent transgenic lines were generated before selecting a stable line.

### Diet preference assay

For diet preference assays, individual CeMbio bacterium (test bacterium) and *E. coli* (control bacterium) were grown in LB broth overnight at 25 °C and adjusted to an $OD_{600}$ of 1. Standard diet choice assays were conducted on 90 mm NGM plates, wherein 25 µl of cultures of the test and control bacteria were spotted 1 cm away from the periphery of the plate on diametrically opposite points of the dish. These plates were incubated at 25 °C for 12 hr. Gravid adult worms were washed thrice with S-basal buffer (100 mM NaCl, 5.75 mM $K_2HPO_4$, 44 mM $KH_2PO_4$). 50–70 worms were placed in the

center of the assay plate and incubated at 25 °C. The preference index (PI) was calculated after 3 hr using the following equation:

$$\text{Preference Index (PI)} = \frac{(\text{Worms on the test lawn}) - (\text{Worms on the control lawn})}{\text{Total worms}}$$

'Odor-only' diet preference assays were performed in tripartitioned plates, wherein one compartment was filled with buffered agar or BA (1 mM $CaCl_2$, 1 mM $MgSO_4$, 5 µg/ml cholesterol, 25 mM $KPO_4$ buffer, and 2% agar) and the other two compartments were filled with NGM. All the assay plates were air-dried for 70–90 min. For EAA supplementation, one of the NGM-filled compartments was supplemented with individual EAA at 5 mM concentration (+EAA), while the other was left unsupplemented (-EAA). All bacterial strains were grown in LB broth overnight at 25 °C and adjusted to an $OD_{600}$ of 1. A volume of 25 µl of the test bacterial culture was spotted on the two compartments with NGM. The plates were incubated at 25 °C for 12 hr. At the time of assay, 1 µl of sodium azide was spotted near the edges of the BA-containing compartment. Gravid adult worms were washed thrice with S-basal buffer, and 50–70 worms were placed at the center of the BA-containing compartment and incubated at 25 °C (schematic in *Figure 1A*). The preference index (PI) was calculated after 3 hr using the following equation:

$$\text{Preference index (PI)} = \frac{[(\text{Worms near diet+EAA}) - (\text{Worms near diet-EAA})]}{\text{Total worms}}$$

To understand worms' preference for individual CeMbio bacterium (test bacterium) over *E. coli* OP50 (control bacterium), we performed an 'odor-only' diet preference assay as described above without any supplementations. The test bacterium culture was spotted in one of the NGM-filled compartments, while *E. coli* OP50 was spotted in the other compartment (schematic in *Figure 1E*). The remaining steps of the assay were performed in a manner similar to the one mentioned above. The PI was calculated using the following equation:

$$\text{Preference Index (PI)} = \frac{(\text{Worms near CeMbio} - \text{Worms near OP50})}{\text{Total worms}}$$

All diet choice experiments were conducted as standard preference assays unless otherwise specified.

## Identification of volatiles produced by microbiota of *C. elegans*

The volatile profiles of CeMbio strains and *E. coli* OP50 were determined using gas chromatography-mass spectrometry (GC-MS/MS) (*Prakash et al., 2021*). Briefly, the individual bacterium was inoculated in 3 ml of LB broth and grown overnight at 25 °C. NGM dishes (60 mm) were seeded with seven spots of 50 µl bacterial cultures adjusted to an $OD_{600}$ of 1. Seeded plates were then incubated at 25 °C for 23 hr. Two plates of the same conditions were sealed together using parafilm and incubated at 25 °C for 1 hr to build odor concentration in the headspace for sampling. The volatiles were collected using a solid-phase microextraction fiber for 1 hr (SPME DVB: Divinylbenzene – Carboxen-WR – Polydimethylsiloxane, 80 µm: Agilent Technologies, Part no. 5191–5874). Odors were subjected to thermal desorption and were identified using gas chromatography-mass spectrometry in an 8890 C gas chromatography machine coupled with a 7000D GC/TQ, which used a capillary column HP-5MS ultra inert (30 m x 0.25 mm and 0.25 m, Agilent 19091S-433UI: 0245625 H) with Helium as the carrier gas at a constant flow rate of 1.5 ml/min. The injection of the sample was done at 40 °C. The GC program included a 40 °C hold for 1 min followed by a temperature ramp to 170 °C at a rate of 5 °C/min, then by a ramp to 270 °C at a rate of 100 °C/min, and finally, a hold for 2 min. The inlet, M.S. source, and M.S. quadrupole temperatures were maintained at 225 °C, 230 °C, and 150 °C, respectively. Odors were then identified using NIST (National Institute of Standards and Technology) 2019 V2.3 Mass Spectral Library and Agilent MassHunter Workstation version 10.0. The m/z peaks detected in the odor profile of NGM plates seeded with LB broth were subtracted from the odor profile of each tested bacterium. For leucine supplementation, preferred diets (CEent1, JUb66, and BIGb0170) were seeded on NGM supplemented with 5 mM leucine (+LEU) and without supplementation (-LEU). GC-MS/MS was performed as described above, and a heat map was generated by analyzing the area under the curve of individual volatile peaks (*Figure 2C*).

For quantification of IAA, a standard curve was prepared by measuring the area under the curve for five concentrations (0.18 µM, 0.36 µM, 2.25 µM, 4.50 µM, and 8.99 µM) of IAA (*Figure 2—figure supplement 1E*). The preferred diets were seeded with two spots of 50 µl culture on NGM dishes supplemented with (+LEU) and without (-LEU) leucine. The volatiles were collected by exposing the fibre for 20 min. The amount of IAA produced by each bacterium was determined by comparing the area under the curve in the GC-MS/MS plot of each bacterium against the standard curve for IAA.

## Bacterial RNA extraction and qRT-PCR relative expression analysis

For RNA isolation, bacterial culture was grown in 20 ml of NGM broth with 5 mM leucine (+LEU) and without leucine (-LEU) for 24 hr at 25 °C. Bacterial cells were harvested by centrifugation at 5000 rpm for 20 min. The bacterial pellet was treated with RNAprotect Bacteria Reagent (Qiagen, Cat. No. 76506). Total RNA was extracted from the samples using the RNeasy Mini kit (Qiagen, Cat. No. 74104), and DNase I (NEB Cat. No. M0303S) treatment was done to remove genomic DNA. cDNA was synthesized using the iScript cDNA synthesis kit (Bio-Rad, Cat. No. 170–8891) and used for qRT-PCR to analyze the relative gene expression of *ilvE* using SYBR Green detection mix (Bio-Rad Cat. No. 1725124) on StepOnePlus (Applied Biosystems) machine. *rboB* was used as a housekeeping gene. The comparative ΔΔCt method was used to determine the fold change of the *ilvE* target gene.

## Chemotaxis assay

Chemotaxis assays were performed as described previously (*Prakash et al., 2021*). Briefly, BA plates (90 mm) air-dried for 90 min were used for the assays. Sodium azide (2 µl) was spotted on the two diametrically opposite ends of the plate, followed by 2 µl of the test chemical to one side and solvent to the opposite side (schematic in Figure S4A). Test chemicals were diluted using suitable solvents (*Supplementary file 5*). Gravid adult worms were washed thrice with S-basal buffer. 50–80 gravid adult worms were placed in the center of the plate and incubated at 25 °C for 3 hr. The chemotaxis index (CI) was calculated using the following equation:

$$\text{Chemotaxis Index (CI)} = \frac{(\text{Worms towards test} - \text{Worms towards control})}{\text{Total worms}}$$

For leucine, we performed a modified chemotaxis assay adapted as described (*Shingai et al., 2005*; *Ward, 1973*). Briefly, 2 µl of leucine solution was spotted on one side of the assay plate and water on the opposite side. The plates were incubated at room temperature for 5 hr to create a leucine gradient. At the time of assay, sodium azide (2 µl) was spotted on the two diametrically opposite ends of the plate. The worms were prepared and introduced on the assay plate in a manner similar to the one mentioned above.

## Odor adaptation

Gravid adult worms were washed thrice with S-basal buffer and transferred to 60 mm NGM plates. For odor adaptation, worms were exposed to bacterial odors by sealing an NGM plate seeded with bacteria (seven spots of 50 µL bacterial culture) with the plate containing washed worms. These plates were incubated at 25 °C for 90 min to desensitize the worms to the odor. After exposure, worms were washed once with S basal buffer and used to perform chemotaxis assays (schematic in *Figure 4A*). For the modified odor adaptation assays, the NGM plate containing worms was exposed to IAA (1:10 dilution) by sealing it with another NGM agar plate containing four spots of 3 µl IAA or without IAA (schematic in *Figure 4—figure supplement 1A*). The adapted worms were then used for diet preference or chemotaxis assays as required.

## Worm movement track analysis

We recorded the motion of five worms on a chemotaxis assay plate using a simple imaging setup. We used 1800 frames (1 fps) to track the behavior of worms. Using MATLAB, we extracted the set of binary images from the recorded frames and processed them using the Trackmate plugin of Fiji software to track worms' motion (*Source code 1*). We used the Advanced Kalman algorithm to create tracks and interpolate for missing frames. The trajectory information was exported in the motilitylab spreadsheet format. This data was analyzed using MATLAB to measure the average speed of worms

and the total distance traveled by worms (*Source code 2*; *Saxton and Jacobson, 1997*; *Codling et al., 2008*; *Li et al., 2008*; *Berg, 1993*).

## Microscopy imaging

A synchronised population of adult worms were used for imaging. Animals were placed in 1 mM Sodium Azide on 2% agarose pads and imaged at 40 X magnification. Imaging was done using a Zeiss LSM880 confocal microscope.

## Statistical analysis

All statistical analyses were done using GraphPad Prism version 8. Statistical analyses were performed either by a two-tailed unpaired $t$-test between two groups or one-way ANOVA, followed by post hoc Dunnett's multiple comparison test across multiple groups. The significant differences were denoted according to p-values: *$p{\leq}0.05$; **$p{\leq}0.01$; ***$p{\leq}0.001$; and ****$p{\leq}0.0001$. Data are presented as means ± SEM. All experiments were performed with 15 biological replicates done over three days.

# Acknowledgements

Some *C. elegans* strains were provided by CGC, which is funded by the NIH Office of Infrastructure Programs. We thank Ms Aatira and Mr Karthick for helping with chemotaxis assays and Ms. Navjot Kaur for some GC-MS analysis. This work was supported by funds from the Wellcome Trust DBT India Alliance, from Indo-French Centre for Advanced Scientific Research, the Royal Society, UK and University of Dundee.

# Additional information

### Funding

| Funder | Grant reference number | Author |
| --- | --- | --- |
| Wellcome Trust DBT India Alliance | IA/S/21/1/505655 | Varsha Singh |
| Indo French Centre for Advanced Scientific Research | IFCP/6503-4 | Marie-Anne Félix Varsha Singh |
| Royal Society | RSWF\R1\231005 | Varsha Singh |

The funders had no role in study design, data collection and interpretation, or the decision to submit the work for publication. For the purpose of Open Access, the authors have applied a CC BY public copyright license to any Author Accepted Manuscript version arising from this submission.

### Author contributions

Ritika Siddiqui, Conceptualization, Resources, Validation, Investigation, Methodology, Writing – original draft, Writing – review and editing; Nikita Mehta, Conceptualization, Validation, Investigation, Methodology, Writing – original draft; Gopika Ranjith, Conceptualization, Investigation; Marie-Anne Félix, Resources, Supervision; Changchun Chen, Resources, Supervision, Methodology; Varsha Singh, Conceptualization, Supervision, Funding acquisition, Writing – original draft, Writing – review and editing

### Author ORCIDs

Ritika Siddiqui ⓘ https://orcid.org/0009-0003-6530-0506
Nikita Mehta ⓘ https://orcid.org/0000-0003-2940-4845
Gopika Ranjith ⓘ https://orcid.org/0000-0002-9149-7037
Changchun Chen ⓘ https://orcid.org/0000-0003-2233-8996
Varsha Singh ⓘ https://orcid.org/0000-0001-9391-5901

Reviewer #1 (Public review): https://doi.org/10.7554/eLife.101936.3.sa1

Reviewer #2 (Public review): https://doi.org/10.7554/eLife.101936.3.sa2

Author response https://doi.org/10.7554/eLife.101936.3.sa3

## Additional files

### Supplementary files

Supplementary file 1. Bacterial strains used in this study.

Supplementary file 2. List of volatiles produced by bacteria.

Supplementary file 3. List of *C. elegans* used in this study.

Supplementary file 4. Details of primers used in this study.

Supplementary file 5. List of volatiles and their respective solvents used for chemotaxis assay.

MDAR checklist

Source code 1. MATLAB code used for extracting the set of binary images from the video.

Source code 2. MATLAB code used for calculating average speed and distance traveled by worms using motility lab spreadsheet.

Source data 1. Source data for all the figures showing the processed values.

### Data availability

Source data has been provided.

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
