## [Editor Report · eLife assessment]

This **important** work is the first to suggest a model that the nematode *C. elegans* prefers specific bacteria (its major food source) that release high amounts of the known attractant isoamyl alcohol when supplemented with exogenous leucine and has also identified a likely receptor for the odorant isoamyl alcohol. The evidence supporting the claims of the authors is **solid**, and the manuscript would be improved by changes to the text that clarify and address the distinction between "supplemented" versus "enriched". The renaming of srd-12 to snif-1 should also be addressed.

---

## [Referee Report · Reviewer #1 (Public review)]

Summary:

Siddiqui et al., investigate the question of how bacterial metabolism contributes to the attraction of *C. elegans* to specific bacteria. They show that *C. elegans* prefers three bacterial species when cultured in a leucine-enriched environment. These bacterial species release more isoamyl alcohol, a known *C. elegans* attractant, when cultured with leucine supplement than without leucine supplement. The study shows correlative evidence that isoamyl alcohol is produced from leucine by the Ehrlich pathway. In addition, they show that SNIF-1 is a receptor for isoamyl alcohol because a null mutant of this receptor exhibits lower chemotaxis to isoamyl alcohol and that chemotaxis to isoamyl alcohol is rescued by expression of snif-1 in AWC.

Strengths:

(1) This study takes a creative approach to examine the question of what specific volatile chemicals released by bacteria may signify to *C. elegans* by examining both bacterial metabolism and *C. elegans* preference behavior. Although *C. elegans* has long been known to be attracted to bacterial metabolites, this study may be one of the first to examine the possible role of a specific bacterial metabolic pathway in mediating attraction.

(2) A strength of the paper is the identification of SNIF-1 as a receptor for isoamyl alcohol. The ligands for very few olfactory receptors have been identified in *C. elegans* and so this is a significant addition to the field. The SNIF-1 null mutant strain will likely be a useful reagent for many labs examining olfactory and foraging behaviors.

Weaknesses:

(1) The authors write that the leucine metabolism via the Ehrlich pathway is required for production of isoamyl alcohol by three bacteria (CEent1, JUb66, BIGb0170), but their evidence for this is correlation and not causation. They show that the gene, ilvE (which is part of the Ehrlich pathway) is upregulated in CEent1 bacteria upon exposure to leucine. Although this indicates that the ilvE gene may be involved in leucine metabolism, it does not show causation. To show causation, they need to knockout ilvE from one of these strains, show that the bacteria does not have increased isoamyl alcohol production when cultured on leucine, and that the bacteria is no longer attractive to *C. elegans*.

(2) Although the authors do show that the three bacterial strains they focus on (CEent1, JUb66, and BIGb0170) are more attractive to *C. elegans* when supplemented with leucine. Some other strains such as BIGb0393 are also more attractive with leucine supplementation and produce isoamyl alcohol (Fig 1B and Supp Table 2). It is unclear why these other strains are not included with the selected three strains.

(3) The behavioral evidence that snif-1 gene encodes a receptor for isoamyl alcohol is compelling because of the mutant phenotype and rescue experiments. The evidence would be stronger with calcium imaging of AWC neurons in response to isoamyl alcohol in the receptor mutant with the expectation that the response would be reduced or abolished in the mutant compared to wildtype.

---

## [Referee Report · Reviewer #2 (Public review)]

Summary:

Siddiqui et al. show that *C. elegans* prefers certain bacterial strains that have been supplemented with the essential amino acid (EEA) leucine. They convincingly show that some leucine enriched bacteria stimulate the production of isoamyl alcohol (IAA). IAA is an attractive odorant that is sensed by the AWC. The authors an identify a receptor, SRD-12, that is expressed in the AWC chemosensory neurons and is required for chemotaxis to IAA. The authors propose that IAA is a predominant olfactory cue that determines diet preference in *C. elegans*. Since leucine is an EAA, the authors propose that worm IAA sensing allows the animal provides a proxy mechanism to identify EAA rich diets.

Strengths:

The authors propose IAA as a predominant olfactory cue that determines diet preference in *C. elegans* providing molecular mechanism underlying diet selection. They show that wild isolates of *C. elegans* have strong chemotactic response to IAA indicating that IAA is an ecologically relevant odor for the worm. The paper is well written, and the presented data are convincing and well organized. This is an interesting paper that connects chemotactic response with bacterially produced odors and thus provides an understanding how animals adapt their foraging behavior through the perception of molecules that may indicate the nutritional value.

Weaknesses:

Major: While I do like the way the authors frame *C. elegans* IAA sensing as mechanisms to identify leucine (EAA) rich diets, it is not fully clear whether bacterial IAA production is a proxy for bacterial leucine levels.

(1) Can the authors measure leucine (or other EAA) content of the different CeMbio strains? This would substantiate the premise in the way they frame this in the introduction. While the authors convincingly show that leucine supplementation induces IAA production in some strains, it is not clear if there are lower leucine levels in the different in the non-preferred strains.

(2) It is not clear whether the non-preferred bacteria in Figure 1A and 1B have the ability to produce IAA. To substantiate the claim that *C. elegans* prefers CEent1, JUb66, and BIGb0170 due to their ability to generate IAA from leucine, it would be measure IAA levels in non-preferred bacteria (+ and - leucine supplementation). If the authors have these data it would be good to include this.

(3) The authors would strengthen their claim if they could show that deletion or silencing ilvE enzyme reduces IAA levels and eliminates the increased preference upon leucine supplementation.

(4) While the three preferred bacteria possess the ilvE gene, it is not clear whether this enzyme is present in the other non-preferred bacterial strains. As far as I know, the CeMbio strains have been sequenced, so it should be easy to determine if the non-preferred bacteria possess the capacity to make IAA. Does expression of ilvE in e.g. *E. coli* increase its preference index or are the other genes in the biosynthesis pathway missing?

(5) It is strongly implied that leucine rich diets are beneficial to the worm. Do the authors have data to show the effect on leucine supplementation on *C. elegans* healthspan, life-span or broodsize?

Comments on revisions:

(1) The authors have addressed most of the earlier questions. The main unresolved issue is the link between iaa production is a reflection of bacterial leucine levels. It is not clear if there are lower leucine levels in the different in non-preferred strains.

The main conclusions that: 1. some bacterial strains can convert exogenous leucine into IAA which is an attractant to *C. elegans*. 2. The identification of a GPCR required for IAA responses are solid. These are important results that carry the paper. My outstanding concern remains with the overinterpretation of the framing that *C. elegans* IAA sensing is used as a mechanism to identify leucine (EAA) rich diets. It is fine to leave this a favorite hypothesis in the discussion but statements throughout the paper need to be nuanced without leucine measurement of the different bacterial strains. (Also since for the bacterial chemotaxis assays there were only done with a single concentration of leucine makes it difficult to infer bacterial leucine concentrations). I recommend softening claims related to leucine-rich diet detection unless quantitative measurements are provided.

Part of the issue in the text lies in the difference between "supplemented" and "chemotaxis" (lab based constructs) and enriched and foraging (natural environment based). This is also the way it is set up in the introduction "Do animals use specific sensing mechanisms to find an EAA-enriched diet?". If enriched is used strictly the same as supplemented then it would be fine but in the text this distinction gets blurred and enriched drifts to the more ethological explanation.

Then it is more than just semantics since leucine-supplemented diets are not something that occurs in the natural environment. IAA production by bacteria could be a signal for a leucine rich environment and it is fine to speculate about this in the discussion.

Examples where the wording needs to be more precise to reflect the experimental results rather than the possible impact in its natural environment:

The title:' The olfactory receptor SNIF-1 mediates foraging for leucine-rich diets in *C. elegans*"

The intro:"Taken together, SNIF-1 regulates the dietary preference of worms to IAA-producing bacteria and thereby mediates the foraging behavior of *C. elegans* to leucine-enriched diets. Thus, IAA produced by bacteria is a dietary quality code for leucine-enriched bacteria."

Results "Figure 1. *C. elegans* relies on odors to select leucine-enriched bacteria"

Supplementation is used more in the text and the figure legends whereas headings and abstract use enriched. The experiments in the paper only describe leucine-supplemented experiments. So use I would supplemented instead of enriched when describing experiments for clarity.

For instance:

Page 4:"Microbial odors drive the preference of *C. elegans* for leucine-enriched diet"

Page 5: "Altogether, these findings suggested that worms rely on odors to distinguish various bacteria and find leucine-enriched bacteria"

Page 7: "Isoamyl alcohol odor is a signature for a leucine-enriched diet"

Page 9: AWC odor sensory neurons facilitate the diet preference of *C. elegans* for leucine-enriched diets"

page 20 "Leucine-enriched diets produce significantly higher levels of IAA odor, making up to 90% of their headspace"

(2) As suggested in the first round of review the authors now add data IAA levels in non-preferred bacteria (+ and - leucine supplementation) in table S2. While it is good to have this data, the table is not very clear. Not clear what ND stands for in the table S2. Not determined or not detected? I assume not determined since some strains Jub44, BiGb0393 Jub134 produce IAA even in the absence of LEU. The authors mention that "the abundance of IAA in these strains is significantly less". However, the table just reflects yes or no. Can the authors give an indication of the concentration to understand what significantly less means? Fig. 2c at least gives a heat map.

(3) On wormbase the gene is still called srd-12. The authors should seek permission to rename srd-12 to snif-1.

---

## [Author Response]

The following is the authors’ response to the original reviews.

**eLife Assessment:**
This is an important study, supported by solid to convincing data, that suggests a model for diet selection in *C. elegans*. The significance is that while *C. elegans* has long been known to be attracted to bacterial volatiles, what specific bacterial volatiles may signify to *C. elegans* is largely unknown. This study also provides evidence for a possible odorant/GPCR pairing.
**Public Reviews:**

**Reviewer #1 (Public review):**
Summary:Siddiqui et al., investigate the question of how bacterial metabolism contributes to the attraction of *C. elegans* to specific bacteria. They show that *C. elegans* prefers three bacterial species when cultured in a leucine-enriched environment. These bacterial species release more isoamyl alcohol, a known *C. elegans* attractant, when cultured with leucine supplement than without leucine supplement. The study shows correlative evidence that isoamyl alcohol is produced from leucine by the Ehrlich pathway. In addition, they show that SRD-12 (SNIF-1) is likely a receptor for isoamyl alcohol because a null mutant of this receptor exhibits lower chemotaxis to isoamyl alcohol and lower preference for leucine-enriched bacteria.Strengths:(1) This study takes a creative approach to examine the question of what specific volatile chemicals released by bacteria may signify to *C. elegans* by examining both bacterial metabolism and *C. elegans* preference behavior. Although *C. elegans* has long been known to be attracted to bacterial metabolites, this study may be one of the first to examine the role of a specific bacterial metabolic pathway in mediating attraction.(2) A strength of the paper is the identification of SRD-12 (SNIF-1) as a likely receptor for isoamyl alcohol. The ligands for very few olfactory receptors have been identified in *C. elegans* and so this is a significant addition to the field. The srd-12 (snif-1) null mutant strain will likely be a useful reagent for many labs examining olfactory and foraging behaviors.Weaknesses:(1) The authors write that the leucine metabolism via the Ehrlich pathway is required for the production of isoamyl alcohol by three bacteria (CEent1, JUb66, BIGb0170), but their evidence for this is correlation and not causation. They write that the gene ilvE is a bacterial homolog of the first gene in the yeast Ehrlich pathway (it would be good to include a citation for this) and that the gene is present in these three bacterial strains. In addition, they show that this gene, ilvE, is upregulated in CEent1 bacteria upon exposure to leucine. To show causation, they need to knockout ilvE from one of these strains, show that the bacteria does not have increased isoamyl alcohol production when cultured on leucine, and that the bacteria is no longer attractive to *C. elegans*.

Thank you for the comment. We have added the appropriate citation [1,2]. We agree that worms’ diet preference for the preferred strains upon ilvE knockout will further strengthen the claim for IAA being used as a proxy for leucine-enriched diet. Currently, protocols and tools for genetic manipulations for CeMbio strains are not available, making this experiment not feasible at this time.

(2) The authors examine three bacterial strains that *C. elegans* showed increased preference when grown with leucine supplementation vs. without leucine supplementation. However, there also appears to be a strong preference for another strain, JUb0393, when grown on plus leucine (Figure 1B). It would be good to include statistics and criteria for selecting the three strains.

Thanks for your comment. We agree that for Pantoea nemavictus, JUb393, worms seem to prefer the leucine supplemented (+ LEU) bacteria over unsupplemented (-LEU). However, when given a choice between the individual CeMbio bacteria and *E. coli* OP50, worms showed preference for only CEent1, JUb66, and BIGb0170 (Figure 1F). Consequently, CEent1, JUb66, and BIGb0170 were selected for further analyses. We have included statistics for Figure 1B-C and Figure S1A-G with details mentioned in the figure legend.

(3) Although the behavioral evidence that srd-12 (snif-1) gene encodes a receptor for isoamyl alcohol is compelling, it does not meet the standard for showing that it is an olfactory receptor in *C. elegans*. To show it is indeed a likely receptor one or more of the following should be done:(a) Calcium imaging of AWC neurons in response to isoamyl alcohol in the receptor mutant with the expectation that the response would be reduced or abolished in the mutant compared to wildtype.(b)"A receptor swap" experiment where the SRD-12 (SNIF-1) receptor is expressed in AWB repulsive neuron in SRD-12 (SNIF-1) receptor mutant background with the expectation that with receptor swap *C. elegans* will now be repulsed from isoamyl alcohol in chemotaxis assays (experiment from Sengupta et al., 1996 odr-10 paper).

Thanks for all your comments and suggestions. While the lab currently does not have the necessary expertise to conduct calcium imaging of neurons, we have performed additional experiments to confirm the requirements of AWC neurons for SNIF-1 function. We generated transgenic worms with extrachromosomal array expressing snif-1 under (a) AWC-specific promoter, odr-1, and (b) AWB-specific promoter, str-1. As shown in new panel 6H in the revised manuscript and Author response image 1, we found that overexpression of snif-1 in AWC neurons completely rescues the chemotaxis defect of snif-1 mutant (referred at VSL2401), whereas upon the “receptor swap" in AWB neurons IAA is sensed as a repellent.

**Author response image 1. sa3fig1:** (A) Chemotaxis index (CI) of WT, VSL2401, VSL2401 [AWCp::snif-1] and VSL2401 [AWBp::snif-1] worms to IAA at 1:1000 dilution. Significant differences are indicated as **** P ≤ 0.0001 determined by one-way ANOVA followed by post hoc Dunnett’s multiple comparison test. Error bars indicate SEM (n≥15).

(4) The authors conclude that *C. elegans* cannot detect leucine in chemotaxis assays. It is important to add the method for how leucine chemotaxis assay was done in order to interpret these results. Because leucine is not volatile if leucine is put on the plates immediately before the worms are added (as in a traditional odor chemotaxis assay), there is no leucine gradient for the worm to detect. It would be good to put leucine on the plate several hours before worms are introduced so worms have the possibility to be able to detect the gradient of leucine (for example, see Wakabayashi et al., 2009).

Previously, the chemotaxis assays with leucine were performed like traditional odor chemotaxis assays. We also performed chemotaxis assay as detailed in Shingai et al 2005[3]. Leucine was spotted on the assay plates 5 hours prior to the introduction of worms on the plates. As shown in new panel S1H in the revised manuscript, wild-type worms do not show response to leucine in the modified chemotaxis assay.

We have included the experimental details for leucine chemotaxis assays in the revised manuscript.

(5) The bacterial preference assay entitled "odor-only assay" is a misleading name. In the assay, *C. elegans* is exposed to both volatile chemicals (odors) and non-volatile chemicals because the bacteria are grown on the assay plate for 12 hours before the worms are introduced to the assay plate. In that time, the bacteria is likely releasing non-volatile metabolites into the plate which may affect the worm's preference. A true odor-only assay would have the bacteria on the lid and the worms on the plate.

The ‘odor-only’ diet preference assay does not allow for non-volatile chemicals to reach worms. We achieved this by using tripartite dishes where the compartments containing worms and bacterial odors are separated by polystyrene barriers. At the time of the assay, worms were spotted in a separate compartment from that of bacteria (as shown in schematic 1A). The soluble metabolites released by the bacteria during their growth will accumulate in the agar within the bacterial compartment alone such that worms only encounter the volatile metabolites produced by bacteria wafting past the polystyrene barrier.

(6) The findings of the study should be discussed more in the context of prior literature. For example, AWC neurons have been previously shown to be involved in bacterial preference (Harris et al., 2014; Worthy et al., 2018). In addition, CeMbio bacterial strains (the strains examined in this study) have been previously shown to release isoamyl alcohol (Chai et al. 2024).

Thanks for the suggestion. We have modified the Discussion section to discuss the study in the light of relevant prior literature.

**Reviewer #2 (Public review):**
Summary:Siddiqui et al. show that *C. elegans* prefers certain bacterial strains that have been supplemented with the essential amino acid (EEA) leucine. They convincingly show that some leucine enriched bacteria stimulate the production of isoamyl alcohol (IAA). IAA is an attractive odorant that is sensed by the AWC. The authors an identify a receptor, SRD-12 (SNIF-1), that is expressed in the AWC chemosensory neurons and is required for chemotaxis to IAA. The authors propose that IAA is a predominant olfactory cue that determines diet preference in *C. elegans*. Since leucine is an EAA, the authors propose that worm IAA sensing allows the animal provides a proxy mechanism to identify EAA rich diets.Strengths:The authors propose IAA as a predominant olfactory cue that determines diet preference in *C. elegans* providing molecular mechanism underlying diet selection. They show that wild isolates of *C. elegans* have a strong chemotactic response to IAA indicating that IAA is an ecologically relevant odor for the worm. The paper is well written, and the presented data are convincing and well organized. This is an interesting paper that connects chemotactic response with bacterially produced odors and thus provides an understanding of how animals adapt their foraging behavior through the perception of molecules that may indicate the nutritional value.Weaknesses:Major:While I do like the way the authors frame *C. elegans* IAA sensing as mechanisms to identify leucine (EAA) rich diets it is not fully clear whether bacterial IAA production is a proxy for bacterial leucine levels.(1) Can the authors measure leucine (or other EAA) content of the different CeMbio strains? This would substantiate the premise in the way they frame this in the introduction. While the authors convincingly show that leucine supplementation induces IAA production in some strains, it is not clear if there are lower leucine levels in the different in non-preferred strains.

Thanks for your suggestion. Estimating leucine levels in various bacteria will provide useful information, and we hope to do so in future studies.

(2) It is not clear whether the non-preferred bacteria in Figure 1A and 1B have the ability to produce IAA. To substantiate the claim that *C. elegans* prefers CEent1, JUb66, and BIGb0170 due to their ability to generate IAA from leucine, it would measure IAA levels in non-preferred bacteria (+ and - leucine supplementation). If the authors have these data it would be good to include this.

Thanks for the suggestion. We have included the table indicating the presence or absence of IAA production by all the bacteria under + LEU and – LEU conditions (Table S2). Some of the nonpreferred bacteria indeed produce isoamyl alcohol. However, the abundance of IAA in these strains is significantly less than in the preferred bacteria.

Using the available genomic sequence data, we found that all CeMbio strains encode IlvE-like transaminase enzymes[4]. This suggests that presumably all the bacteria have the metabolic capacity to make alpha-ketoisocaproate (an intermediate in IAA biosynthetic pathway) from leucine. However, the regulation of metabolic flux is likely to be quite complex in various bacteria.

(3) The authors would strengthen their claim if they could show that deletion or silencing ilvE enzyme reduces IAA levels and eliminates the increased preference upon leucine supplementation.

We agree that testing worms’ diet preference for the preferred strains upon ilvE knockout will further strengthen the claim for IAA being crucial for finding leucine-enriched diet. Currently the lab does not have the necessary expertise and standardize protocols to do genetic manipulations for the CeMbio strains.

(4) While the three preferred bacteria possess the ilvE gene, it is not clear whether this enzyme is present in the other non-preferred bacterial strains. As far as I know, the CeMbio strains have been sequenced so it should be easy to determine if the non-preferred bacteria possess the capacity to make IAA. Does the expression of ilvE in e.g. *E. coli* increase its preference index or are the other genes in the biosynthesis pathway missing?

Thanks for the suggestion. Using the available genomic sequence data, we find that all the bacteria in the CeMbio collection possess IlvE-like transaminase necessary for synthesis of alphaketoisocaproate, key metabolite in leucine turn over as well as precursor for IAA [4]. *E. coli* has an IlvE encoding gene in its genome [2]. However, we do not find IAA in the headspace of *E. coli* either with or without leucine supplementation. This indicates either (i) *E. coli* lacks enzymes for subsequent steps in IAA biosynthesis or (ii) leucine provided under the experimental regime is not sufficient to shift the metabolic flux to IAA production.

Previous studies have suggested that in yeast, the final two steps leading to IAA production are catalyzed by decarboxylase and dehydrogenase enzymes1. The genomic and metabolic flux data available for CeMbio do not describe specific enzymes leading up to IAA synthesis [4].

(5) It is strongly implied that leucine-rich diets are beneficial to the worm. Do the authors have data to show the effect on leucine supplementation on *C. elegans* healthspan, life-span or broodsize?

Edwards et al. 2015 reported a 15% increase in the lifespan of worms upon 1 mM leucine supplementation [5]. Wang et al 2018 also showed lifespan extension upon 1 mM and 10 mM leucine supplementation. They also reported that while leucine supplementation did not have any effect on brood size, it did make worms more resistant to heat, paraquat, and UV-stress [6]. These studies have been included in the discussion section.

Other comments:Page 6. Figure 2c. While the authors' conclusions are correct based on AWC expts. it would be good at this stage to include the possibility that odors that enriched in the absence of leucine may be aversive.

Thanks for the comment. We have tested the chemotaxis response of the worms for most of the odors produced by CeMbio strains without leucine supplementation. We did not find any odor that is aversive to worms. However, we cannot completely rule out the possibility that a low abundance of aversive odor in the headspace of the bacteria was missed.

Interestingly, we did identify 2-nonanone, a known repellent, in the headspace of the preferred bacteria upon leucine supplementation. However, the abundance of 2-nonanone in headspace of bacteria is relatively low (less than 1% for CEent1, and JUb66, and ~10% for BIGb0170). This suggests that the relative abundance of odors in an odor bouquet may be a relevant factor in determining worms’ reference.

Page 6. IAA increases 1.2-4 folds upon leucine supplementation. If the authors perform a chemotaxis assay with just IAA with 1-2-4 fold differences do you get the shift in preference index as seen with the bacteria? i.e. is the difference in IAA concentration sufficient to explain the shift in bacterial PI upon leucine supplementation? Other attractants such as Acetoin and isobutanol go up in -Leu conditions.

Thanks for the suggestion. As shown in Figure S2H and S2I, when given a choice between a concentration of IAA (1:1000 dilution) attractive to worms and a 4-fold higher amount of IAA, worms chose the latter. This result suggests that worms can distinguish between relatively small difference in concentrations of IAA.

We agree that the relative abundance of Acetoin and Isobutanol is high in -LEU conditions. The presence of other attractants in - LEU conditions should skew the preference of worms for – LEU bacteria. However, we found that worms prefer + LEU bacteria (Figure 1B), suggesting that the abundance of IAA mainly influences the diet preference of the worms.

Page 14-15. The authors identify a putative IAA receptor based on expression studies. I compliment the authors for isolating two CRISPR deletion alleles. They show that the srd-12 (snif-1) mutants have obvious defects in IAA chemotaxis. Very few ligand-odorant receptors combinations have been identified so this is an important discovery. CenGen data indicate that srd-12 (snif-1) is expressed in a limited set of neurons. Did the authors generate a reporter to show the expression of srd-12 (snif-1)? This is a simple experiment that would add to the characterization of the SRD-12 (SNIF-1) receptor. Rescue experiments would be nice even though the authors have independent alleles. To truly claim that SRD-12 (SNIF-1) is the ligand for IAA and activates the AWC neurons would require GCamp experiments in the AWC neuron or heterologous expression system. I understand that GCamp imaging might not be part of the regular arsenal of the lab but it would be a great addition (even in collaboration with one of the many labs that do this regularly). Comparing AWC activity using GCaMP in response IAA-producing bacteria with high leucine levels in both wild-type and SRD-12 (SNIF-1) deficient backgrounds, would further support their narrative. I leave that to the authors.

Thanks for your comments and suggestions. To address this comment, we rescued snif-1 mutant (referred as VSL2401) with extrachromosomal array expressing snif-1 under AWC-specific promoter as well as its native promoter. As shown in Figure 6H and Author response image 2, we find that both transgenic lines show a complete rescue of chemotaxis response to isoamyl alcohol. To find where snif-1 is expressed, we generated a transgenic line of worms expressing GFP under snif-1 promoter, and mCherry under odr-1 promoter (to mark AWC neurons). As shown in Figure 6I, we found that snif-1 is expressed faintly in many neurons, with strong expression in one of the two AWC neurons marked by odr-1::mCherry. This result suggests that SNIF-1 is expressed in AWC neuron.

We hope to perform GCaMP assay and further characterization of SNIF-1 in the future.

**Author response image 2. sa3fig2:** Chemotaxis index (CI) of WT, VSL2401, VSL2401 [AWCp:: snif-1] and VSL2401 [snif-1p::snif-1] worms to IAA at 1:1000 dilution. Significant differences are indicated as **** P ≤ 0.0001 determined by one-way ANOVA followed by post hoc Dunnett’s multiple comparison test. Error bars indicate SEM (n≥15).

Minor:Page 4 "These results suggested that worms can forage for diets enriched in specific EAA, leucine...." More precise at this stage would be to state " These results indicated that worms can forage for diets supplemented with specific EAA...".

We have changed the statement in the revised manuscript.

Page 5."these findings suggested that worms not only rely on odors to choose between two bacteria but also to find leucine enriched bacteria" This statement is not clear to me and doesn't follow the data in Fig. S2. Preferred diets in odorant assays are the IAA producing strains.

Thanks for your comment. We have revised the manuscript to make it clear. “Altogether, these findings suggested that worms rely on odors to distinguish different bacteria and find leucineenriched bacteria”. This statement concludes all the data shown in Figure 1 and Figure S1.

Page 5. Figure S2A provides nice and useful data that can be part of the main Figure 1.

Thanks for the comment. We have incorporated the data from Figure S2A to main Figure 1.

**Reviewer #3 (Public review):**
Summary:The authors first tested whether EAA supplementation increases olfactory preference for bacterial food for a variety of bacterial strains. Of the EAAs, they found only leucine supplementation increased olfactory preference (within a bacterial strain), and only for 3 of the bacterial strains tested. Leucine itself was not found to be intrinsically attractive.They determined that leucine supplementation increases isoamyl alcohol (IAA) production in the 3 preferred bacterial strains. They identify the biochemical pathway that catabolizes leucine to IAA, showing that a required enzyme for this pathway is upregulated upon supplementation.Consistent with earlier studies, they find that AWC olfactory neuron is primarily responsible for increased preference for IAA-producing bacteria.Testing volatile compounds produced by bacteria and identified by GC/MS, and identified several as attractive, most of them require AWC for the full effect. Adaptation assays were used to show that odorant levels produced by bacterial lawns were sufficient to induce olfactory adaptation, and adaptation to IAA reduced chemotaxis to leucine-supplemented lawns. They then showed that IAA attractiveness is conserved across wild strains, while other compounds are more variable, suggesting IAA is a principal foraging cue.Finally, using the CeNGEN database, they developed a list of candidate IAA receptors. Using behavioral tests, they show that mutation of srd-12 (snif-1) greatly impairs IAA chemotaxis without affecting locomotion or attraction to another AWC-sensed odor, PEA.CommentsThis study will be of great interest in the field of *C. elegans* behavior, chemical senses and chemical ecology, and understanding of the sensory biology of foraging.Strengths:The identification of a receptor for IAA is an excellent finding. The combination of microbial metabolic chemistry and the use of natural bacteria and nematode strains makes an extremely compelling case for the ecological and adaptive relevance of the findings.Weaknesses:AWC receives synaptic input from other chemosensory neurons, and thus could potentially mediate navigation behaviors to compounds detected in whole or in part by those neurons. Language concluding detection by AWC should be moderated (e.g. p9 "worms sense an extensive repertoire...predominantly using AWC") unless it has been demonstrated.

Thanks for your comment. We have modified the manuscript to incorporate the suggestion.

srd-12 (snif-1) is not exclusively expressed in AWC. Normally, cell-specific rescue or knockdown would be used to demonstrate function in a specific cell. The authors should provide such a demonstration or explain why they are confident srd-12 (snif-1) acts in AWC.

Thanks for the comment. We have performed AWC-specific rescue of snif-1 in mutant worms. As shown in Figure 6H, we found that AWC neurons specific rescue completely recovered the chemotaxis defect of the snif-1 mutant (referred as VSL2401) for IAA. In addition, snif-1 is expressed in one of the AWC neurons.

A comparison of AWC's physiological responses between WT and srd-12 (snif-1), preferably in an unc13 background, would be nice. Even further, the expression of srd-12 (snif-1) in a different neuron type and showing that it confers responsiveness to IAA (in this case, inhibition) would be very convincing.

Thanks for the suggestion. We have performed a receptor swap experiment, where snif-1 is misexpressed in AWB neurons. We find that these worms show slight but significant repulsion to IAA compared to WT and snif-1 mutant worms (Author response image 1).

**Recommendations for the authors:**

**Reviewing Editor:**
Please consider all of the reviewer comments. In particular, as noted in the individual reviews, the strength of the evidence would be bolstered by additional experiments to demonstrate that the iLvE enzyme affects IAA levels in the preferred bacteria. The reviewers note that the authors haven't shown that IAA production is a reflection of leucine content. Are the non-preferred bacteria low on leucine or lack iLvE or IAA synthesis pathways? Further, more direct evidence that SRD-12 (SNIF-1) is in fact the primary IAA receptor would further strengthen the study. The authors should also be aware that geographic distance for wild isolate *C. elegans* may not directly correlate with phylogenetic distance. This should be assessed/discussed for the strains used.

Thanks for the suggestions. Some of these have been addressed in response to reviewers. Thanks for your comments about possible disconnect between geographical and phylogenetic distances amongst natural isolates used here.

By analyzing the phylogenetic tree generated using neighbor-joining algorithm available at CaeNDR database, we found that QX1211 and JU3226 are phylogenetically close, but the remaining isolates fall under different clades separated by long phylogenetic distances [7,8].

**Reviewer #1 (Recommendations for the authors):**
(1) In the first sentence of the third paragraph of the introduction, *C. elegans* are described as "soildwelling." Although *C. elegans* has been described as soil-dwelling in the past, current research indicates they are most often found on rotten fruit, compost heaps and other bacterial-rich environments, not soil. "All Caenorhabditis species are colonizers of nutrient- and bacteria-rich substrates and none of them is a true soil nematode." from Kiontke, K. and Sudhaus, W. Ecology of Caenorhabditis species (WormBook).

Your specific comment about *C. elegans*’ habitat is well received. However, **i**n that sentence we are referring to the chemosensory system of soil-dwelling animals in general, and not particularly *C. elegans*.

(2) Figure 3K, the model would be clearer if leucine-rich diet -> volatile chemicals ->AWC (instead of leucine-rich diet -> AWC <- volatile chemicals). The leucine-rich diet results in the production of volatile chemicals which are detected by AWC.

We have modified the figure to make it clearer.

(3) Figure 4 - it would help to include a table summarizing the volatile chemicals that each bacteria releases. Then the reader could more easily evaluate whether the adaptation to each specific odor is consistent with the change in preference for the specific bacteria based on what it releases in its headspace. In addition, Figure 4 would help to clarify whether bacteria in these experiments were cultured with or without leucine supplementation.

Table S2 summarizes the odors released by all the bacteria under + LEU and – LEU conditions.

In Figure 4, adaptation was performed by odors of bacteria when cultured under leucineunsupplemented conditions.

**Reviewer #2 (Recommendations for the authors):**
Page 9. Previous studies e.g. Bargmann Hartwieg and Horvitz have shown IAA is sensed by the AWC. Would be good to cite appropriately.

Thanks for the comment. The reference has been cited at p9 and p16.

References:

(1) Yuan, J., Mishra, P., and Ching, C.B. (2017). Engineering the leucine biosynthetic pathway for isoamyl alcohol overproduction in *Saccharomyces cerevisiae*. Journal of Industrial Microbiology and Biotechnology 44, 107-117. 10.1007/s10295-016-1855-2 %J Journal of Industrial Microbiology and Biotechnology.

(2) Kanehisa, M., Furumichi, M., Sato, Y., Matsuura, Y., and Ishiguro-Watanabe, M. (2025). KEGG: biological systems database as a model of the real world. Nucleic Acids Res 53, D672-d677. 10.1093/nar/gkae909.

(3) Shingai, R., Wakabayashi, T., Sakata, K., and Matsuura, T. (2005). Chemotaxis of *Caenorhabditis elegans* during simultaneous presentation of two water-soluble attractants, llysine and chloride ions. Comparative biochemistry and physiology. Part A, Molecular & integrative physiology 142, 308-317. 10.1016/j.cbpa.2005.07.010.

(4) Dirksen, P., Assié, A., Zimmermann, J., Zhang, F., Tietje, A.M., Marsh, S.A., Félix, M.A., Shapira, M., Kaleta, C., Schulenburg, H., and Samuel, B.S. (2020). CeMbio - The *Caenorhabditis elegans* Microbiome Resource. G3 (Bethesda, Md.) 10, 3025-3039. 10.1534/g3.120.401309.

(5) Edwards, C., Canfield, J., Copes, N., Brito, A., Rehan, M., Lipps, D., Brunquell, J., Westerheide, S.D., and Bradshaw, P.C. (2015). Mechanisms of amino acid-mediated lifespan extension in *Caenorhabditis elegans*. BMC genetics 16, 8. 10.1186/s12863-015-0167-2.

(6) Wang, H., Wang, J., Zhang, Z.J.J.o.F., and Research, N. (2018). Leucine Exerts Lifespan Extension and Improvement in Three Types of Stress Resistance (Thermotolerance, AntiOxidation and Anti-UV Irradiation) in *C. elegans*. 6, 665-673.

(7) Crombie, T.A., McKeown, R., Moya, N.D., Evans, Kathryn S., Widmayer, Samuel J., LaGrassa, V., Roman, N., Tursunova, O., Zhang, G., Gibson, Sophia B., et al. (2023). CaeNDR, the Caenorhabditis Natural Diversity Resource. Nucleic Acids Research 52, D850-D858. 10.1093/nar/gkad887 %J Nucleic Acids Research.

(8) Cook, D.E., Zdraljevic, S., Roberts, J.P., and Andersen, E.C. (2017). CeNDR, the *Caenorhabditis elegans* natural diversity resource. Nucleic Acids Res 45, D650-d657. 10.1093/nar/gkw893.